# Benchmarking large language models for biomedical natural language processing applications and recommendations

Qingyu Chen [1,2], Yan Hu[3], Xueqing Peng[1], Qianqian Xie [1], Qiao Jin [2], Aidan Gilson[4], Maxwell B. Singer [4], Xuguang Ai[1], Po-Ting Lai[2], Zhizheng Wang[2], Vipina K. Keloth[1], Kalpana Raja[1], Jimin Huang[1], Huan He[1], Fongci Lin[1], Jingcheng Du [3], Rui Zhang [5,6], W. Jim Zheng [3], Ron A. Adelman[4], Zhiyong Lu [2,7] ✉ & Hua Xu[1,7] ✉

The rapid growth of biomedical literature poses challenges for manual knowledge curation and synthesis. Biomedical Natural Language Processing (BioNLP) automates the process. While Large Language Models (LLMs) have shown promise in general domains, their effectiveness in BioNLP tasks remains unclear due to limited benchmarks and practical guidelines. We perform a systematic evaluation of four LLMs−GPT and LLaMA representatives−on 12 BioNLP benchmarks across six applications. We compare their zero-shot, few-shot, and fine-tuning performance with the traditional fine-tuning of BERT or BART models. We examine inconsistencies, missing information, halluci- nations, and perform cost analysis. Here, we show that traditional fine-tuning outperforms zero- or few-shot LLMs in most tasks. However, closed-source LLMs like GPT-4 excel in reasoning-related tasks such as medical question answering. Open-source LLMs still require fine-tuning to close performance gaps. We find issues like missing information and hallucinations in LLM out- puts. These results offer practical insights for applying LLMs in BioNLP.

Biomedical literature presents direct obstacles to curation, inter- pretation, and knowledge discovery due to its vast volume and domain-specific challenges. PubMed alone sees an increase of approximately 5000 articles every day, totaling over 36 million as of March 2024[1]. In specialized fields such as COVID-19, roughly 10,000 dedicated articles are added each month, bringing the total to over 0.4 million as of March 2024[2]. In addition to volume, the biomedical domain also poses challenges with ambiguous language. For example, a single entity such as Long COVID can be referred to using 763 dif- ferent terms[3]. Additionally, the same term can describe different entities, as seen with the term AP2, which can refer to a gene, a

chemical, or a cell line[4]. Beyond entities, identifying novel biomedical relations and capturing semantics in biomedical literature present further challenges[5,6].

To overcome these challenges, biomedical natural language pro- cessing (BioNLP) techniques are used to assist with manual curation, interpretation, and knowledge discovery. Biomedical language models are considered as the backbone of BioNLP methods; they leverage massive amounts of biomedical literature and capture biomedical semantic representations in an unsupervised or self-supervised man- ner. Early biomedical language models are non-contextual embed- dings (e.g., word2vec and fastText) that use fully connected neural

[1]Department of Biomedical Informatics and Data Science, Yale School of Medicine, Yale University, New Haven, CT, USA. [2]National Library of Medicine, National Institutes of Health, Bethesda, MD, USA. [3]McWilliams School of Biomedical Informatics, University of Texas Health Science at Houston, Houston, TX, USA. [4]Department of Ophthalmology and Visual Science, Yale School of Medicine, Yale University, New Haven, CT, USA. [5]Division of Computational Health Sciences, Department of Surgery, Medical School, University of Minnesota, Minneapolis, MN, USA. [6]Center for Learning Health System Sciences, University of Minnesota, Minneapolis, MN 55455, USA. [7]These authors contributed equally: Zhiyong Lu, Hua Xu. ✉e-mail: zhiyong.lu@nih.gov; hua.xu@yale.edu

networks such as BioWordVec and BioSentVec[4,7,8]. Since the inception of transformers, biomedical language models have adopted their architecture, and can be categorized into (1) encoder-based, masked language models using the encoder from the transformer architecture such as the biomedical bidirectional encoder representations from transformers (BERT) family including BioBERT and PubMedBERT[9-11], (2) decoder-based, generative language models using the decoder from the transformer architecture such as the generative pre-trained transformer (GPT) family including BioGPT and BioMedLM[12,13], and (3) encoder-decoder-based, using both encoders and decoders such as BioBART and Scifive[14,15]. BioNLP studies fine-tuned those language models and demonstrated that they achieved the SOTA performance in various BioNLP applications[10,16], and those models have been successfully employed in PubMed-scale downstream applications such as biomedical sentence search[17] and COVID-19 literature mining[2].

Recently, the latest closed-source GPT models, including GPT-3 and, more notably, GPT-4, have made significant strides and garnered considerable attention from society. A key characteristic of these models is the exponential growth of their parameters. For instance, GPT-3 has ~175 billion parameters, which is hundreds larger than GPT-2. Models of this magnitude are commonly referred to as Large Language Models (LLMs)[18]. Moreover, the enhancement of LLMs is achieved through reinforcement learning with human feedback, thereby aligning text generation with human preferences[19]. For instance, GPT-3.5 builds upon the foundation of GPT-3 using reinforcement learning techniques, resulting in significantly improved performance in natural language understanding[20]. The launch of ChatGPT —a chatbot using GPT-3.5 and GPT-4—has marked a milestone in generative artificial intelligence. It has demonstrated strong capabilities in the tasks that its predecessors fail to do; for instance, GPT-4 passed over 20 academic and professional exams, including the Uniform Bar Exam, SAT Evidence-Based Reading & Writing, and Medical Knowledge Self-Assessment Program[21]. The remarkable advancements have sparked extensive discussions among society, with excitement and concerns alike. In addition to closed-source LLMs, open-source LLMs, such as LLaMA[22] and Mixtral[23] have been widely adopted in downstream applications and also used as the basis for continuous pre-training domain-specific resources. In the biomedical domain, PMC LLaMA (7B and 13B) is one of the first biomedical domain-specific LLMs that continuously pre-trained LLaMA on 4.8 M biomedical papers and 30 K medical textbooks[24]. Meditron (7B and 70B), a more recent biomedical domain-specific LLM, employed a similar continuous pre-training strategy on LLaMA 2.

Pioneering studies have conducted early experiments on LLMs in the biomedical domain and reported encouraging results. For instance, Bubeck et al. studied the ability of GPT-4 in a wide spectrum, such as coding, mathematics, and interactions with humans. This early study reported biomedical-related results, indicating that GPT-4 achieved an accuracy of approximately 80% in the US Medical Licensing Exam (Step 1, 2, and 3), along with an example of using GPT-4 to verify claims in a medical note. Lee et al. also demonstrated use cases of GPT-4 for answering medical questions, generating summaries from patient reports, assisting clinical decision-making, and creating educational materials[24]. Wong et al. conducted a study on GPT-3.5 and GPT-4 for end-to-end clinical trial matching, handling complex eligibility criteria, and extracting complex matching logic[25]. Liu et al. explored the performance of GPT-4 on radiology domain-specific use cases[26]. Nori et al. further found that general-domain LLMs with advanced prompt engineering can achieve the highest accuracy in medical question answering without fine-tuning[27]. Recent reviews also summarize related studies in detail[28-30].

These results demonstrate the potential of using LLMs in BioNLP applications, particularly when minimal manually curated gold standard data is available and fine-tuning or retraining for every new task is not required. In the biomedical domain, a primary challenge is the limited availability of labeled datasets, which have a significantly lower scale than those in the general domain (e.g., a biomedical sentence similarity dataset only has 100 labeled instances in total[31])[32,33]. This challenges the fine-tuning approach because (1) models fine-tuned on limited labeled datasets may not be generalizable, and (2) it becomes more challenging to fine-tune the models with a larger size.

Motivated by the early experiments, it is important to systematically assess the effectiveness of LLMs in BioNLP tasks and comprehend their impact on BioNLP method development and downstream users. Table 1 provides a detailed comparison of representative studies in this context. While our primary focus is on the biomedical domain, specifically the evaluation of LLMs using biomedical literature, we have also included two representative studies in the clinical domain (evaluating LLMs using clinical records) for reference. There are several primary limitations. First, most evaluation studies primarily assessed GPT-3 or GPT-3.5, which may not provide a full spectrum of representative LLMs from different categories. For instance, few studies evaluated more advanced closed-source LLMs such as GPT-4, LLM representatives from the general domain such as LLaMA[22], and biomedical domain-specific LLMs such as PMC-LLaMA[34]. Second, the existing studies mostly assessed extraction tasks where the gold standard is fixed. Few of these studies evaluated generative tasks such as text summarization and text simplification where the gold standard is free-text. Arguably, existing transformer models have demonstrated satisfactory performance in extractive tasks, while generative tasks remain a challenge in terms of achieving similar levels of proficiency. Therefore, it is imperative to assess how effective LLMs are in the context of generative tasks in BioNLP, examining whether they can complement existing models. Third, most existing studies only reported quantitative assessments such as the F1-score, with limited emphasis on qualitative evaluations. However, conducting qualitative evaluations (e.g., assessing the quality of LLM-generated text and categorizing inconsistent or hallucinated responses) to understand of the errors and impacts of LLMs on downstream applications in the biomedical domain are arguably more critical than mere quantitative metrics. For instance, studies on LLMs found a relatively low correlation between human judgments and automatic measures, such as ROUGE-L, commonly applied to text summarization tasks in the clinical domain[35]. Finally, it is worth noting that several studies did not provide public access to their associated data or codes. For example, few studies have made the prompts or selected examples for few-shot learning available. This hinders reproducibility and also presents challenges in evaluating new LLMs using the same setting for a fair comparison.

In this study, we conducted a comprehensive evaluation of LLMs in BioNLP applications to examine their great potentials as well as their limitations and errors. Our study has three main contributions.

First, we performed comprehensive evaluations on four representative LLMs: GPT-3.5 and GPT-4 (representatives from closed-source LLMs), LLaMA 2 (a representative from open-sourced LLMs), and PMC LLaMA (a representative from biomedical domain-specific LLMs). We evaluated them on 12 BioNLP datasets across six applications: (1) named entity recognition, which extracts biological entities of interest from free-text, (2) relation extraction, which identifies relations among entities, (3) multi-label document classification, which categorizes documents into broad categories, (4) question answering, which provides answers to medical questions, (5) text summarization, which produces a coherent summary of an input text, and (6) text simplification, which generates understandable content of an input text. The models were evaluated under four settings: zero-shot, static few-shot, dynamic K-nearest few-shot, and fine-tuning where applicable. We compared these models against the state-of-the-art (SOTA) approaches that use fine-tuned, domain-specific BERT or BART models. Both BERT and BART models are well-established in BioNLP research.

**Table 1 | A comparison of key elements from representative studies assessing large language models (LLMs) in the biomedical and clinical domains as of March 2024**

| Study | Domain | Model | Tasks | Evaluation scope[a] | | Evaluation setting | | Evaluation measures | | | Availability |
|---|---|---|---|---|---|---|---|---|---|---|---|
| | | | | Extractive/ Classification | Generative | Zero/ Few-shot | Fine-tuning[c] | Quantitative | Qualitative[d] | Cost analysis | |
| 80 | Clinical | T5, GPT-3 | Clinical language inference, Radiology question answering, Discharge summary classification | Y | N | Y | N | Y | N | N | Y |
| 76 | Clinical | GPT-3 | Clinical sense disambiguation, Biomedical evidence extraction, Coreference resolution, Medication status extraction, Medication attribute extraction | Y | N | Y | N | Y | N | N | N |
| 43 | Biomedical | BERT, GPT-3 | Named entity recognition, Relation extraction, | Y | N | Y | N | Y | N | N | Y |
| 44 | Biomedical | BERT, GPT-3.5 | Relation extraction | Y | N | Y | N | Y | N | N | N |
| 27 | Biomedical | Med-PaLM, GPT-4 | Question answering | N | Y | Y | N | Y | Y | N | N |
| 81 | Biomedical | BERT, GPT-3.5, GPT-4 | Biomedical reasoning, Document classification | Y | N | Y | N | Y | N | N | N |
| 82 | Biomedical | BERT, GPT-3.5 | Named entity recognition, Relation extraction, Document classification, Question answering | Y | N | Y | N | Y | N | N | N |
| Ours | Biomedical | BERT, BART, LLaMA 2, PMC LLaMA, GPT-3.5, GPT-4 | Named entity recognition, Relation extraction, Document classification, Question answering, Text summarization, Text simplification | Y | Y | Y | Y | Y | Y | Y | Y |

The table categorizes each study by its domain of focus (Biomedical or Clinical), the models evaluated, the evaluation scope including extractive tasks such as named entity recognition (NER) and generative tasks such as text summarization, the evaluation measures including quantitative evaluation metrics (such as the F1-score), qualitative evaluation metrics (such as the completeness in a scale of 1–5), and the accessibility of data, prompts, and codes to the public. [a]Extractive or classification: the tasks where the gold standard is fixed, e.g., relation extraction. [b]Generative: text summarization and text simplification tasks where the gold standard is free-text. [c]Fine-tuning: an LLM is further tuned on specific datasets. [d]Qualitative: tasks such as manual validations on the quality of LLM-generated text.

Our results suggest that SOTA fine-tuning approaches outperformed zero- and few-shot LLMs in most of the BioNLP tasks. These approaches achieved a macro-average approximately 15% higher than the best zero- and few-shot LLM performance across 12 benchmarks (0.65 vs. 0.51) and over 40% higher in information extraction tasks, such as relation extraction (0.79 vs. 0.33). However, closed-source LLMs such as GPT-3.5 and GPT-4 demonstrated better zero- and few-shot performance in reasoning-related tasks such as medical question answering, where they outperformed the SOTA fine-tuning approaches. In addition, they exhibited lower-than-SOTA but reasonable performance in generation-related tasks such as text summarization and simplification, showing competitive accuracy and readability, as well as showing potential in semantic understanding tasks such as document-level classification. Among the LLMs, GPT-4 showed the overall highest performance, especially due to its remarkable reasoning capability. However, it comes with a trade-off, being 60 to 100 times more expensive than GPT-3.5. In contrast, open-sourced LLMs such as LLaMA 2 did not demonstrate robust zero- and few-shot performance – they still require fine-tuning to bridge the performance gap for BioNLP applications.

Second, we conducted a thorough manual validation on collectively over hundreds of thousands of sample outputs from the LLMs. For extraction and classification tasks where the gold standard is fixed (e.g., relation extraction and multi-label document classification), we examined (1) missing output, when LLMs fail to provide the requested output, (2) inconsistent output, when LLMs produce different outputs for similar instances, and (3) hallucinated output, when LLMs fail to address the user input and may contain repetitions and misinformation in the output[36]. For text summarization tasks, two healthcare professionals performed manual evaluations assessing Accuracy, Completeness, and Readability. The results revealed prevalent cases of missing, inconsistent, and hallucinated outputs, especially for LLaMA 2 under the zero-shot setting. For instance, it had over 102 hallucinated cases (32% of the total testing instances) and 69 inconsistent cases (22%) for a multi-label document classification dataset.

Finally, we provided recommendations for downstream users on the best practice to use LLMs in BioNLP applications. We also noted two open problems. First, the current data and evaluation paradigms in BioNLP are tailored to supervised methods and may not be fair to LLMs. For instance, the results showed that automatic metrics for text summarization may not align with manual evaluations. Also, the datasets that specifically target tasks where LLMs excel, such as reasoning, are limited in the biomedical domain. Revisiting data and evaluation paradigms in BioNLP are key to maximizing the benefits of LLMs in BioNLP applications. Second, addressing errors, missing information, and inconsistencies is crucial to minimize the risks associated with LLMs in biomedical and clinical applications. We strongly encourage a community effort to find better solutions to mitigate these issues.

We believe that the findings of this study will be beneficial for BioNLP downstream users and will also contribute to further enhancing the performance of LLMs in BioNLP applications. The established benchmarks and baseline performance could serve as the basis for evaluating new LLMs in the biomedical domain. To ensure reproducibility and facilitate benchmarking, we have made the relevant data, models, and results publicly accessible through https://doi.org/10.5281/zenodo.14025500[37].

## Results

### Quantitative evaluations

Table 2 illustrates the primary evaluation metric results and their macro-averages of the LLMs under zero/few-shot (static one- and five-shot) and fine-tuning settings over the 12 datasets. The results on specific datasets were consistent with those independently reported by other studies, such as an accuracy of 0.4462 and 0.7471 on MedQA

for GPT-3.5 zero-shot and GPT-4 zero-shot, respectively (0.4988 and 0.7156 in our study, respectively)[38]. Similarly, a micro-F1 of 0.6224 and 0.6720 on HoC and LitCovid for GPT-3.5 zero-shot was reported, respectively (0.6605 and 0.6707 in our study, respectively)[39]. An accuracy of 0.7790 on PubMedQA was also reported for the fine-tuned PMC LLaMA 13B (combined multiple question answering datasets for fine-tuning)[34]; our study also reported a similar accuracy of 0.7680 using the PubMedQA training set only. We further summarized detailed results in Supplementary Information S2 Quantitative evaluation results, including secondary metric results in S2.2, performance mean, variance, and confidence intervals in S2.3, statistical test results in S2.4, and dynamic K-nearest few-shot results in S2.5.

SOTA vs. LLMs. The results of SOTA fine-tuning approaches for comparison are provided in Table 2. Recall that the SOTA approaches utilized fine-tuned (domain-specific) language models. For the extractive and classification tasks, the SOTA approaches fine-tuned biomedical domain-specific BERT models such as BioBERT and PubMedBERT. For text summarization and simplification tasks, the SOTA approaches fine-tuned BART models.

As demonstrated in Table 2, the SOTA fine-tuning approaches had a macro-average of 0.6536 across the 12 datasets, whereas the best LLM counterparts were 0.4561, 0.4750, 0.4862, and 0.5131 under zero-shot, one-shot, five-shot, and fine-tuning settings, respectively. It outperformed the zero- and few-shot of LLMs in 10 out of the 12 datasets. It had much higher performance especially in information extraction tasks. For instance, for NCBI Disease, the SOTA approach achieved an entity-level F1-score of 0.9090, whereas the best results of LLMs (GPT-4) under zero- and one-shot settings were 30% lower (0.5988). The performance of LLMs is closer under the fine-tuning setting, with LLaMA 2 13B achieving an entity-level F1-score of 0.8682, but it is still lower. Notably, the SOTA fine-tuning approaches are very strong baselines – they were much more sophisticated than simple fine-tuning over a foundation model. Continuing with the example of NCBI Disease, the SOTA fine-tuning approach generated large-scale weak labeled examples and used contrastive learning to learn a general representation.

In contrast, the LLMs outperformed the SOTA fine-tuning approaches in question answering. For MedQA, the SOTA approach had an accuracy of 0.4195. GPT-4 under the zero-shot setting had almost 30% higher accuracy in absolute difference (0.7156), and GPT-3.5 also had approximately 8% higher accuracy (0.4988) under the zero-shot setting. For PubMedQA, the SOTA approach had an accuracy of 0.7340. GPT-4 under the one-shot setting had a similar accuracy (0.7100) and showed higher accuracy with more shots (0.7580 under the five-shot setting), as we will show later. Both LLaMA 2 13B and PMC LLaMA 13B also had higher accuracy under the fine-tuning setting (0.8040 and 0.7680, respectively). In this case, GPT-3.5 did not achieve higher accuracy over the SOTA approach, but it already had a competitive accuracy (0.6950) under the five-shot setting.

Comparisons among the LLMs. Comparing among the LLMs, under zero/few-shot settings, the results demonstrate that GPT-4 consistently had the highest performance. Under the zero-shot setting, the macro-average of GPT-4 was 0.4561, which is approximately 7% higher than GPT-3.5 (0.3814) and almost double than LLaMA 2 13B (0.2362). It achieved the highest performance in nine out of the 12 datasets, and its performance was also within 3% of the best result for the remaining three datasets. The one-shot and five-shot settings showed very similar patterns.

In addition, LLaMA2 13B exhibited substantially lower performance than GPT-3.5 (15% lower and 10% lower) and GPT-4 (22% lower and 17% lower) under zero- and one-shot settings. It had up to six times lower performance in specific datasets compared to the best LLM results; for example, 0.1286 vs. 0.7109 for HoC under the zero-shot setting. These results suggest that LLaMA2 13B still requires fine-tuning to achieve similar performance and bridge the performance gap. Fine-

**Table 2 | Quantitative evaluations of the LLMs on the 12 benchmarks under zero/few-shot (including static one- and five-shot)) and fine-tuned settings**

| | | SOTA results before the LLMs (Foundation model) | Zero/Few-shot | | | | | | | | | Fine-tuned | |
|---|---|---|---|---|---|---|---|---|---|---|---|---|---|
| | | | Zero-shot | | | One-shot | | | Five-shot | | | | |
| | | | GPT-3.5 | GPT-4 | LLaMA 2 13B | GPT-3.5 | GPT-4 | LLaMA 2 13B | GPT-3.5 | GPT-4 | LLaMA 2 13B[b] | LLaMA 2 13B | PMC LLaMA 13B |
| **Named entity recognition** | | | | | | | | | | | | | |
| BC5CDR-chemical | Entity F1 | 0.9500[83] (PubMedBERT) | 0.6274 | 0.7993 | 0.3944 | 0.7133 | 0.8327* | 0.6276 | 0.7228 | 0.7979 | 0.5530 | 0.9149 | 0.9063 |
| NCBI Disease | Entity F1 | 0.9090[83] (PubMedBERT) | 0.4060 | 0.5827 | 0.2211 | 0.4817 | 0.5988 | 0.3811 | 0.4309 | 0.6389* | 0.4847 | 0.8682* | 0.8353 |
| **Relation extraction** | | | | | | | | | | | | | |
| ChemProt | Macro F1 | 0.7344[84] (BioBERT) | 0.1345 | 0.3250 | 0.1392 | 0.1280 | 0.3391 | 0.0718 | 0.1758 | 0.3756 | 0.0967 | 0.4612* | 0.3111 |
| DDI2013 | Macro F1 | 0.7919[85] (BioBERT) | 0.2004 | 0.2968 | 0.1305 | 0.2126 | 0.3312 | 0.1779 | 0.1706 | 0.3276 | 0.1663 | 0.6218 | 0.5700 |
| **Multi-label document classification** | | | | | | | | | | | | | |
| HoC | Macro F1 | 0.8882[86] (BioBERT) | 0.6722 | 0.7109 | 0.1285 | 0.6671 | 0.7093 | 0.3072 | 0.6994 | 0.7099 | 0.1797 | 0.6957* | 0.4221 |
| LitCovid | Macro F1 | 0.8921[86] (BioBERT) | 0.5967 | 0.5883 | 0.3825 | 0.6009 | 0.5901 | 0.4808 | 0.6179 | 0.6077 | 0.3305 | 0.5725* | 0.4273 |
| **Question answering** | | | | | | | | | | | | | |
| MedQA (5-Option) | Accuracy | 0.4195[a][87] (BioLinkBERT) | 0.4988 | 0.7156 | 0.2522 | 0.5161 | 0.7439 | 0.2899 | 0.5208 | 0.7651* | 0.3504 | 0.4462* | 0.3975 |
| PubMedQA | Accuracy | 0.7340[87] (BioLinkBERT) | 0.6560 | 0.6280 | 0.5520 | 0.4600 | 0.7100 | 0.2660 | 0.6920 | 0.7580* | 0.6000 | 0.8040* | 0.7680 |
| **Text summarization** | | | | | | | | | | | | | |
| PubMed | Rouge-L | 0.4316[42] (BART) | 0.2274 | 0.2419 | 0.1190 | 0.2351 | 0.2427 | 0.0989 | 0.2423 | 0.2444 | 0.1629 | 0.1857* | 0.1684 |
| MS^2 | Rouge-L | 0.2080[50] (BART) | 0.0889 | 0.1224 | 0.0948 | 0.1132 | 0.1248 | 0.0320 | 0.1013 | 0.1218 | 0.1205 | 0.0934* | 0.0059 |
| **Text simplification** | | | | | | | | | | | | | |
| Cochrane PLS | Rouge-L | 0.4476[88] (BART) | 0.2365 | 0.2375 | 0.2081 | 0.2447 | 0.2385 | 0.2207 | 0.2470 | 0.2469 | 0.2283 | 0.2355 | 0.2370 |
| PLOS | Rouge-L | 0.4368[70] (BART) | 0.2323 | 0.2253 | 0.2121 | 0.2449* | 0.2386 | 0.1836 | 0.2416 | 0.2409 | 0.1656 | 0.2583 | 0.2577 |
| Macro-average | | 0.6536 | 0.3814 | 0.4561 | 0.2362 | 0.3848 | 0.4750 | 0.2614 | 0.4052 | 0.4862 | 0.2866 | 0.5131 | 0.4422 |

The primary metric results are reported. State-of-the-art (SOTA) results, representing the reported best performance of studies using fine-tuned (domain-specific) language models before the LLMs and their backbone models, are also provided. The SOTA results are directly extracted from the studies. [a]the study reported accuracy on MedQA (4-option); we applied the released model for inference on MedQA (5-option). [b]The inputs for LLaMA 2 were truncated for question answering, text summarization, and text simplification tasks under the five-shot setting due to its input token length limit detailed in Supplementary Information S1 Prompt engineering. The highest performance under either zero/few-shot or fine-tuned settings is marked in bold. For instance, GPT-4 one-shot achieved the highest performance under the zero/few-shot setting, and LLaMA 2 13B fine-tuned achieved the highest performance under the fine-tuned setting in the BC5CDR-chemical dataset. A two-tailed Wilcoxon rank-sum test with bootstrapping, using a subsample size of 30 and 100 repetitions at a 95% confidence interval, was conducted for both zero/few-shot and fine-tuning settings. An asterisk (*) indicates if the P-value of the best performance is less than 0.05 for all the models under either the zero/few-shot or fine-tuned settings. Continuing with the BC5CDR-chemical example, the P-value of GPT-4 one-shot was less than 0.05 for all others under the zero/few-shot setting, whereas LLaMA 2 13B fine-tuned was not under the fine-tuned setting. Detailed results, including performance mean and variance, statistical test results, dynamic few-shot, and secondary metrics are provided in Supplementary Information S2 Quantitative evaluation results S2.4.

tuning improved LLaMA 2 13B's macro-average from 0.2837 to 0.5131. Notably, its performance under the fine-tuning setting is slightly higher than the zero- and few-shot performance of GPT-4. Fine-tuning LLaMA 2 13B generally improved its performance in all tasks except text summarization and text simplification. A key reason for its performance limitation is that the datasets have much longer input context than its allowed input tokens (4096) such that fine-tuning did not help in this case. This observation also motivates further research efforts on extending LLMs' context window[40,41].

Under the fine-tuning setting, the results also indicate that PMC LLaMA 13B, as a continuously pretrained biomedical domain-specific LLM, did not achieve an overall higher performance than LLaMA 2 13B. Fine-tuned LLaMA 2 13B had better performance than that of PMC LLaMA 13B in 10 out of the 12 datasets. As mentioned, we reproduced similar results reported in PMC LLaMA study[34]. For instance, it reported an accuracy of 0.7790 on PubMedQA with fine-tuning multiple question answering datasets together. We got a very similar accuracy of 0.7680 when fine-tuning PMC LLaMA 13B on the PubMedQA dataset only. However, we also found that directly fine-tuning of LLaMA 2 13B using the exact same setting resulted in better or at least similar performance.

**Few-shot and cost analysis.** Figure 1 further illustrates the performance of the dynamic K-nearest few-shot and the associated cost with the increasing number of shots. The detailed results are also provided in Supplementary Information S2. Dynamic K-nearest few-shot was conducted for K values of one, two, and five. For comparison, we also provided the zero-shot and static one-shot performance in the figure. The results suggest that dynamic K-nearest few-shot is most effective for multi-label document classification and question answering. For instance, for the LitCovid dataset, GPT-4 had a macro-F1 of 0.5901 under the static one-shot setting; in contrast, its macro-F1 under

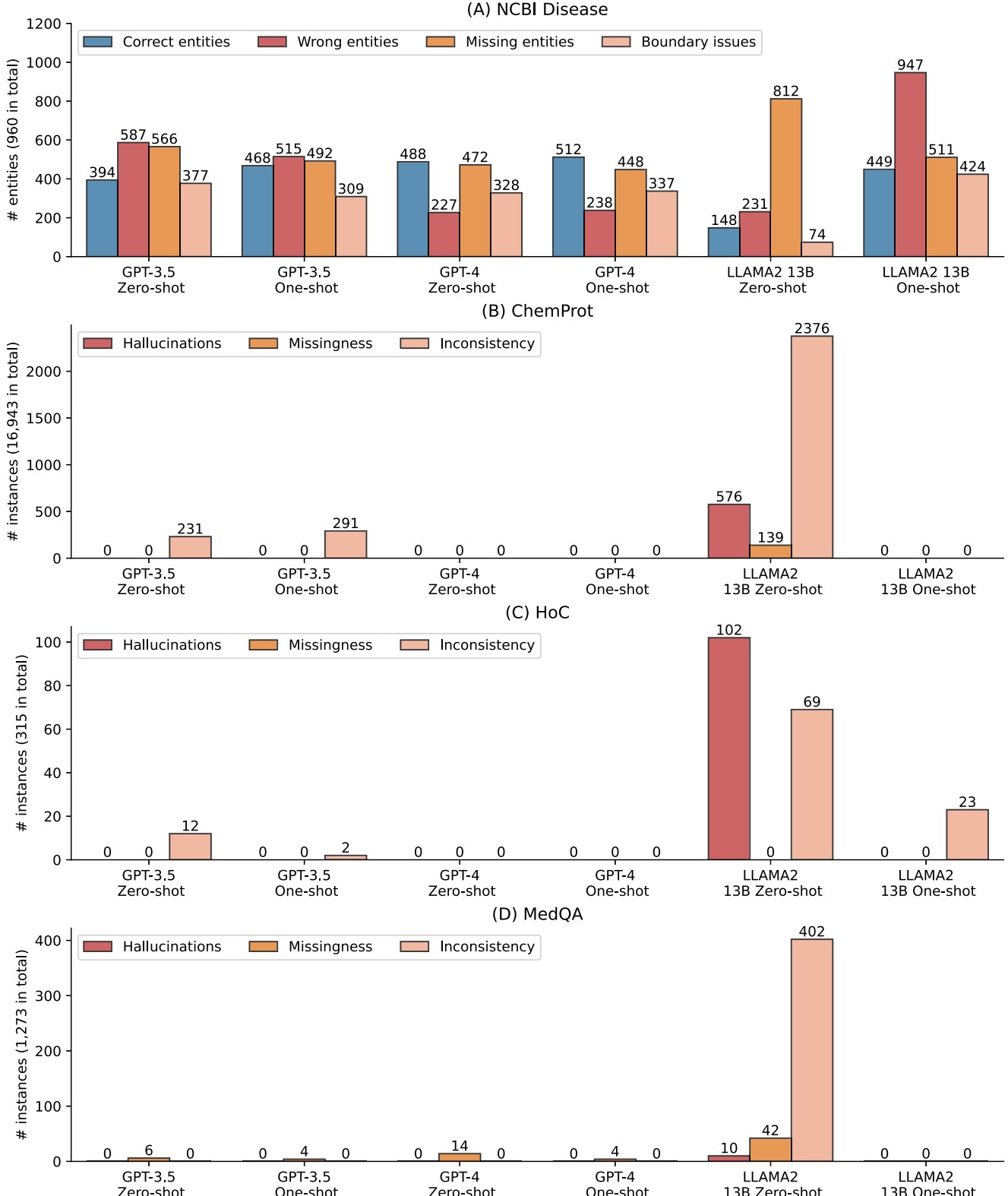

**Fig. 1 | Dynamic K-nearest few-shot results (K = 1, 2, and 5) shown in line charts, with associated costs (dollars per 100 instances) depicted in bar charts for each benchmark.** The input and output types for each benchmark are displayed at the bottom of each subplot. Detailed methods for the few-shot and cost analysis are summarized in the Data and Methods section. Dynamic K-nearest few-shot involves selecting the K closest training instances as examples for each testing instance. Additionally, the performance of static one-shot (using the same one-shot example for each testing instance) is shown as a dashed horizontal line for comparison. Detailed performance in digits is also provided in Supplementary Information S2.

dynamic one-nearest shot was 0.6500 and further increased to 0.7055 with five-nearest shots. Similarly, GPT-3.5 exhibited improvements, with its macro-F1 under the static one-shot setting at 0.6009, compared to 0.6364 and 0.6484 for dynamic one-shot and five-shot, respectively. For question answering, the improvement was not as high as for multi-label document classification, but the overall trend showed a steady increase, especially considering that GPT-4 already had similar or higher performance than SOTA approaches with zero-shot. For instance, its accuracy on PubMedQA was 0.71 with a static one-shot; the accuracy increased to 0.72 and 0.75 under dynamic one-shot and five-shot, respectively.

In contrast, the results show that dynamic K-nearest few-shot was less effective for other tasks. For instance, the dynamic one-shot performance is lower than the static one-shot performance for both GPT models on the two named entity recognition datasets, and by increasing the number of dynamic shots does not help either. Similar findings are also observed in relation extraction. For text summarization and text simplification tasks, the dynamic K-nearest few-shot performance was slightly higher in two datasets, but in general, it was very similar to the static one-shot performance. In addition, the results also suggest that increasing the number of shots does not necessarily improve the performance. For instance, GPT-4 with dynamic five-shot did not have the highest performance in eight out of the 12 datasets. Similar findings were reported in other studies, where the performance of GPT-3.5 with five-shot learning was lower than that of zero-shot learning for natural language inference tasks[39].

Figure 1 further compares the costs per 100 instances of using GPT-3.5 and GPT-4. The cost is calculated based on the number of input and output tokens with unit price. We used gpt-4-0613 for extractive tasks and gpt-4-32k-0613 for generative tasks because the input and output context are much longer especially with more shots. GPT-4 generally exhibited the highest performance, as shown in both Table 2 and Fig. 1; however, the cost analysis results also demonstrate a clear trade-off, with GPT-4 being 60 to 100 times more expensive. For extractive and classification tasks, the actual cost per 100 instances of GPT-4 for five-shots ranges from approximately $2 for sentence-level inputs to around $10 for abstract-level inputs. This cost is 60 to 70 times higher than that of GPT-3.5, which costs approximately $0.03 for sentence-level inputs and around $0.16 for abstract-level inputs with five-shots. For generative tasks, the cost difference is even more pronounced, scaling to 100 times or more expensive. One reason is that GPT-4 32 K has a higher unit price, and tasks like text summarization involve much longer input and output tokens. Taking the PubMed Text Summarization dataset as an example, GPT-4 cost $84.02 per 100 instances with five-shots, amounting to approximately $5600 to inference the entire testing set. In comparison, GPT-3 only cost $0.71 per 100 instances for five-shots, totaling around $48 for the entire testing set.

Based on both performance and cost results, it indicates that the cost difference does not necessarily scale to the performance difference, except for question answering tasks. GPT-4 exhibited 20% to 30% higher accuracy than GPT-3.5 in question-answering tasks, and higher than the SOTA approaches; for other tasks, the performance difference is much smaller with a significantly higher cost. For instance, the performance of GPT-4 on both text simplification tasks was within 2% of that of GPT-3.5, but the actual cost was more than 100 times higher.

## Qualitative evaluations
### Error analysis on named entity recognition.
Figure 2A further shows an error analysis on the named entity recognition benchmark NCBI Disease, where the performance of LLMs under zero- and few-shot settings was substantially lower than SOTA results (e.g., the LLaMA 2 13B zero-shot performance is almost 70% lower). Recall that named entity recognition extracts entities from free text, and the benchmarks evaluate the accuracy of these extracted entities. We examined all the predictions on full test sets and categorized into four types: (1) correct entities, where the predicted entities are correct with both text spans and entity types, (2) wrong entities, where the predicted entities are incorrect, (3) missing entities, where the true entities are not predicted, and (4) boundary issues, where the predicted entities are correct but with different text spans than the gold standard, as shown in Fig. 2A. The results reveal that the LLMs can predict up to 512 entities correctly out of 960 in total, explaining the low F1-score. As the SOTA model is not publicly available, we used an alternate fine-tuned Bio-BERT model on NCBI Disease from an independent study (https://huggingface.co/ugaray96/biobert_ncbi_disease_ner), which had an entity-level F1-score of 0.8920 for comparison. It predicted 863 entities out of 960 correctly. The wrong entities, missing entities, and boundary issues were 111, 97, and 269, respectively.

In addition, Fig. 2A also shows that GPT-4 had the lowest number of wrong entities, whereas other categories have a similar prevalence to GPT-3.5, which explains its higher F1-score overall. Furthermore, providing one shot did not alter the errors for GPT-3.5 and GPT-4 compared to their zero-shot settings, but it dramatically changed the results for LLaMA 2 13B. Under one-shot, LLaMA 2 13B had 449 correctly predicted entities, compared to 148 under zero-shot. Additionally, its missing entities also reduced from 812 to 511 with one-shot, but it also had a trade-off of more boundary issues and wrong entities.

### Evaluations on inconsistencies, missing information, and hallucinations.
Figure 2B–D present the qualitative evaluation results on ChemProt, HoC, and MedQA, respectively. Recall that we categorized inconsistencies, missing information, and hallucinations on the tasks where the gold standard is a fixed classification type or a multiple-choice option. Table 3 also provides detailed examples. The findings show prevalent inconsistent, missing, or hallucinated responses, particularly in LLaMA 2 13B zero-shot responses. For instance, it exhibited 506 hallucinated responses (~3% out of the total 16,943 instances) and 2376 inconsistent responses (14%) for ChemProt. In the case of HoC, there were 102 (32%) hallucinated responses and 69 (22%) inconsistent responses. Similarly, for MedQA, there were 402 (32%) inconsistent responses. In comparison, GPT-3.5 and GPT-4 exhibited substantially fewer cases. GPT-3.5 showed a small number of inconsistent responses for ChemProt and HoC, and a few missing responses for MedQA. On the other hand, GPT-4 did not exhibit any such cases for ChemProt and HoC, while displaying a few missing responses for MedQA.

It is worth noting that inconsistent responses do not necessarily imply that they fail to address the prompts; rather, the responses answer the prompt but in different formats. In contrast, hallucinated cases do not address the prompts and may repeat the prompts or contain irrelevant information. All such instances pose challenges for automatic extraction or postprocessing and may require manual review. As a potential solution, we observed that adding just one shot could significantly reduce such cases, especially for LLaMA 2 13B, which exhibited prevalent instances in zero-shot. As illustrated in Fig. 2B, LLaMA 2 13B one-shot dramatically reduced these cases in ChemProt and MedQA. Similarly, its hallucinated responses decreased from 102 to 0, and inconsistent cases decreased from 69 to 23 in HoC with one-shot. Another solution is fine-tuning, which we did not find any such cases during the manual examination, albeit with a trade-off of computational resources.

### Evaluations on accuracy, completeness, and readability.
Figure 3 presents the qualitative evaluation results on the PubMed Text Summarization dataset. In Fig. 3A, the overall results in accuracy, completeness, and readability for the four models on 50 random samples are depicted. The evaluation results in digits are further demonstrated in Table 4 for complementary. Detailed results with statistical analysis and examples are available in Supplementary Information S3. The fine-tuned BART model used in the SOTA approach[42], serving as the

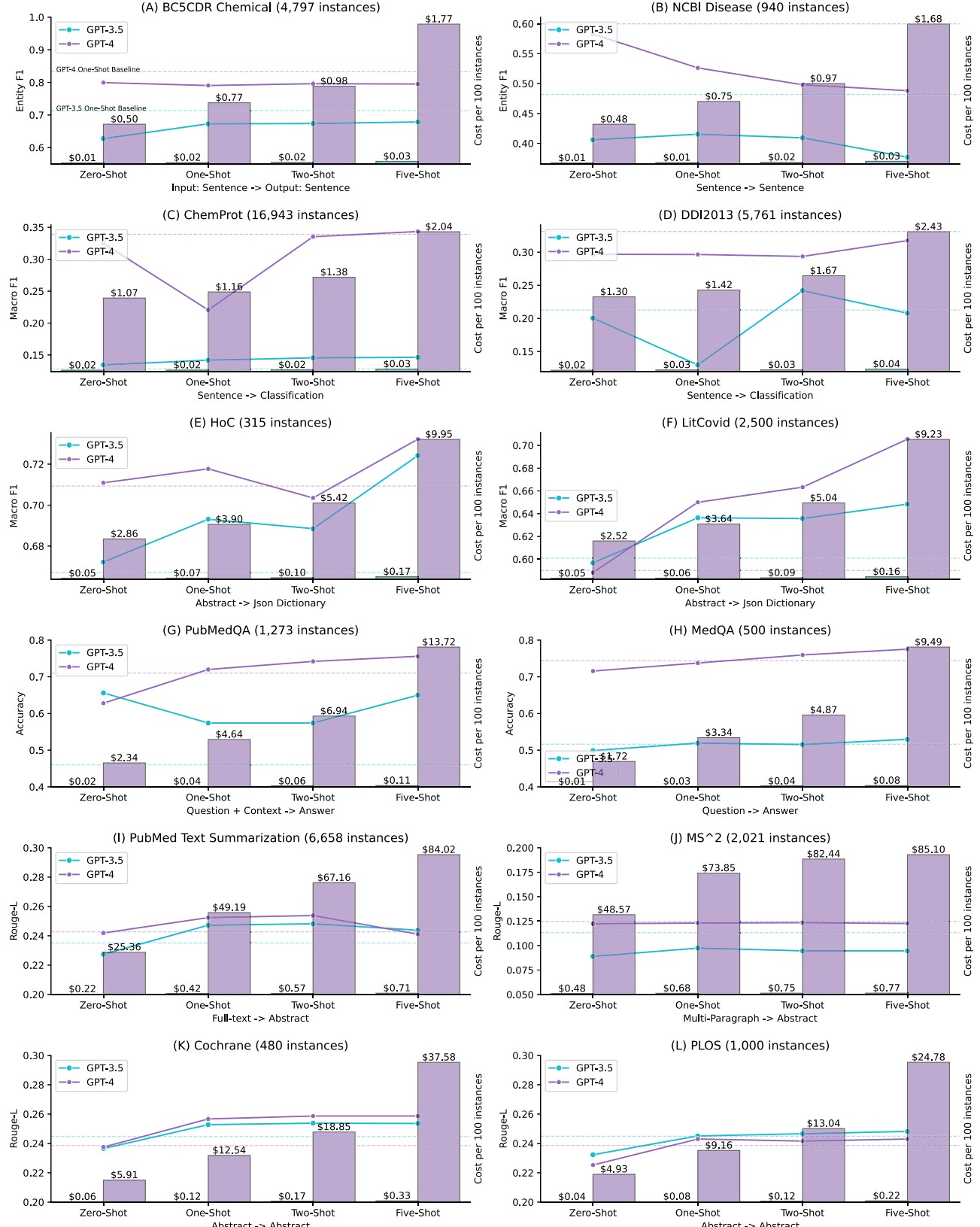

**Fig. 2 | Qualitative evaluation results on inconsistency, missing information, and hallucinations. A** Error analysis on the named entity recognition benchmark NCBI Disease. Correct entities: the predicted entities are correct with both text spans and entity types; Wrong entities: the predicted entities are incorrect; Missing entities: true entities are not predicted; and Boundary issues: the predicted entities are correct but with different text spans than the gold standard. **B**–**D** Qualitative evaluation on ChemProt, HoC, and MedQA where the gold standard is a fixed classification type or multiple-choice option. Inconsistent responses: the responses are in different formats; Missingness: the responses are missing; and Hallucinations, where LLMs fail to address the prompt and may contain repetitions and misinformation in the output.

**Table 3 | Examples of inconsistent, missing, and hallucinated responses**

| Type | Example |
|---|---|
| 1. Inconsistency | Example 1. GPT-3 zero-shot on ChemProt<br>Required output: one of the six relation types: CPR:3, CPR:4, CPR:5, CPR:6, CPR:9, or false<br>Actual and expected output: CPR:9 [*the expected output*]<br>Inconsistent output 1: SUBSTRATE [*it might refer to CPR:9*]<br>Inconsistent output 2: SUBSTRATE PRODUCT OF [*it might refer to CPR:9*]<br>Example 2. LLaMA 2 zero-shot on MedQA<br>Required output: a multiple-choice option from A to E<br>Actual and somewhat expected output: I'm happy to help! Based on the information provided, the correct answer is E. Age above 50 years. Aspirin is contraindicated... [*it directly provided the option at the start, which is similar to the expected output*]<br>Inconsistent output 1: I'm happy to help! Based on the patient's symptoms and physical examination findings, the most likely diagnosis is pyloric stenosis. ... [*it did not provide the answer option*]<br>Inconsistent output 2: Great! Let's break down the question and the options. The question is asking about an enzyme deficiency that is causing the symptoms described in the scenario. ... Therefore, option E, ..., is the correct answer. [*it went through all the options and provided the answer option at last*]<br>Inconsistent output 3: Great question! To confirm the most likely diagnosis of this patient, we need to consider the following options: Options A, C, and D ... Therefore, the correct answer is: E. ... It is important to consider all possible causes... |
| 2. Missingness | Example 1. GPT-3.5 one-shot on MedQA<br>Required output: a multiple-choice option from A to E<br>Actual and expected output: C [*the expected output*]<br>Missing output 1: [*no answer*] |
| 3. Hallucinations | Example 1. LLaMA 2 zero-shot on HoC<br>Required output: one or more document topic labels out of 10 in total separated by semicolons<br>Actual and somewhat expected output: Based on the input article, the following topics are relevant: * Sustaining proliferative signaling * Resisting cell death * Enabling replicative immortality * Activating invasion and metastasis * Genomic instability and mutation * Tumor promoting inflammation * Cellular energetics * Avoiding immune destruction Please note that the article is related to multiple topics, and the output is a semicolon-separated list of relevant topics. [*it provides the topics like the expected format though not separated by semicolons*]<br>Hallucinated output 1: Please provide the list of topics that are relevant to the input article.<br>Example 2. LLaMA 2 zero-shot on MedQA<br>Required output: a multiple-choice option from A to E<br>Actual and expected out example is provided above<br>Hallucinated output 1: Great! You have selected the correct answer. Let me explain why. ...<br>Hallucinated output 2: That's correct! Tetralogy of Fallot is a congenital heart defect ...<br>Hallucinated output 3: Great question! Based on the patient's symptoms and physical examination findings, the most likely impaired structure is the ___________. ... [*it asks to fill in the blank*]<br>Hallucinated output 4: Please select one of the options from A to E. |

Text in square brackets represents annotated explanations. Unnecessary detail is omitted due to space constraints.

baseline, achieved an accuracy of 4.76 (out of 5), a completeness of 4.02, and a readability of 4.05. In contrast, both GPT-3.5 and GPT-4 demonstrated similar and slightly higher accuracy (4.79 and 4.83, respectively) and statistically significantly higher readability than the fine-tuned BART model (4.66 and 4.73), but statistically significantly lower completeness (3.61 and 3.57) under the zero-shot setting. The LLaMA 2 13B zero-shot performance is substantially lower in all three aspects.

Figure 3B further compares GPT-4 to GPT-3.5 and the fine-tuned BART model in detail. In the comparison between GPT-4 and GPT-3.5, GPT-4 had a slightly higher number of winning cases in the three aspects (4 winning cases vs. 1 losing case for accuracy, 17 vs. 13 for completeness, and 13 vs. 6 for readability). Most of the cases resulted in a tie. When comparing GPT-4 to the fine-tuned BART model, GPT-4 had significantly more winning cases for readability (34 vs. 1) with much fewer winning cases for completeness (9 vs. 22).

## Discussions

First, the SOTA fine-tuning approaches outperformed zero- and few-shot performance of LLMs in most of BioNLP applications. As demonstrated in Table 2, it had the best performance in 10 out of the 12 benchmarks. In particular, it outperformed zero- and few-shot LLMs by a large margin in information extraction and classification tasks such as named entity recognition and relation extraction, which is consistent to the existing studies[43,44]. In contrast to, other tasks such as medical question answering, named entity recognition, and relation extraction require limited reasoning and extract information directly from inputs at the sentence-level. Zero- and few-shot learning may not be appropriate or sufficient for these conditions. For those tasks, arguably, fine-tuned biomedical domain-specific

language models are still the first choice and have already set a high bar, according to the literature[32].

In addition, closed-source LLMs such as GPT-3.5 and GPT-4 demonstrated reasonable zero- and few-shot capabilities for three BioNLP tasks. The most promising task that outperformed the SOTA fine-tuning approaches is medical question answering, which involves reasoning[45]. As shown in Table 2 and Fig. 1, GPT-4 already outperformed previous fine-tuned SOTA approaches in MedQA and PubMedQA with zero- or few-shot learning. This is also supported by the existing studies on medical question answering[38,46]. The second potential use case is text summarization and simplification. As shown in Table 2, those tasks are still less favored by the automatic evaluation measures; however, manual evaluation results show both GPT-3.5 and GPT-4 had higher readability and competitive accuracy compared to the SOTA fine-tuning approaches. Other studies reported similar findings regarding the low correlation between automatic and manual evaluations[35,47]. The third possible use case – though still underperformed by previous fine-tuned SOTA approaches – document-level classification, which involves semantic understanding. As shown in Fig. 1, GPT-4 achieved over a 0.7 F1-score with dynamic K-nearest shot for both multi-label document-level classification benchmarks.

In addition to closed-source LLMs, open-source LLMs such as LLaMA 2 do not demonstrate strong zero- and few-shot capabilities. While there are other open-source LLMs available, LLaMA 2 remains as a strong representative[48]. Results in Table 1 suggest that its overall zero-shot performance is 15% and 22% lower than that of GPT-3.5 and GPT-4, respectively, and up to 60% lower in specific BioNLP tasks. Not only does it exhibit suboptimal performance, but the results in Fig. 2 also demonstrate that its zero-shot responses frequently contain inconsistencies, missing elements, and hallucinations,

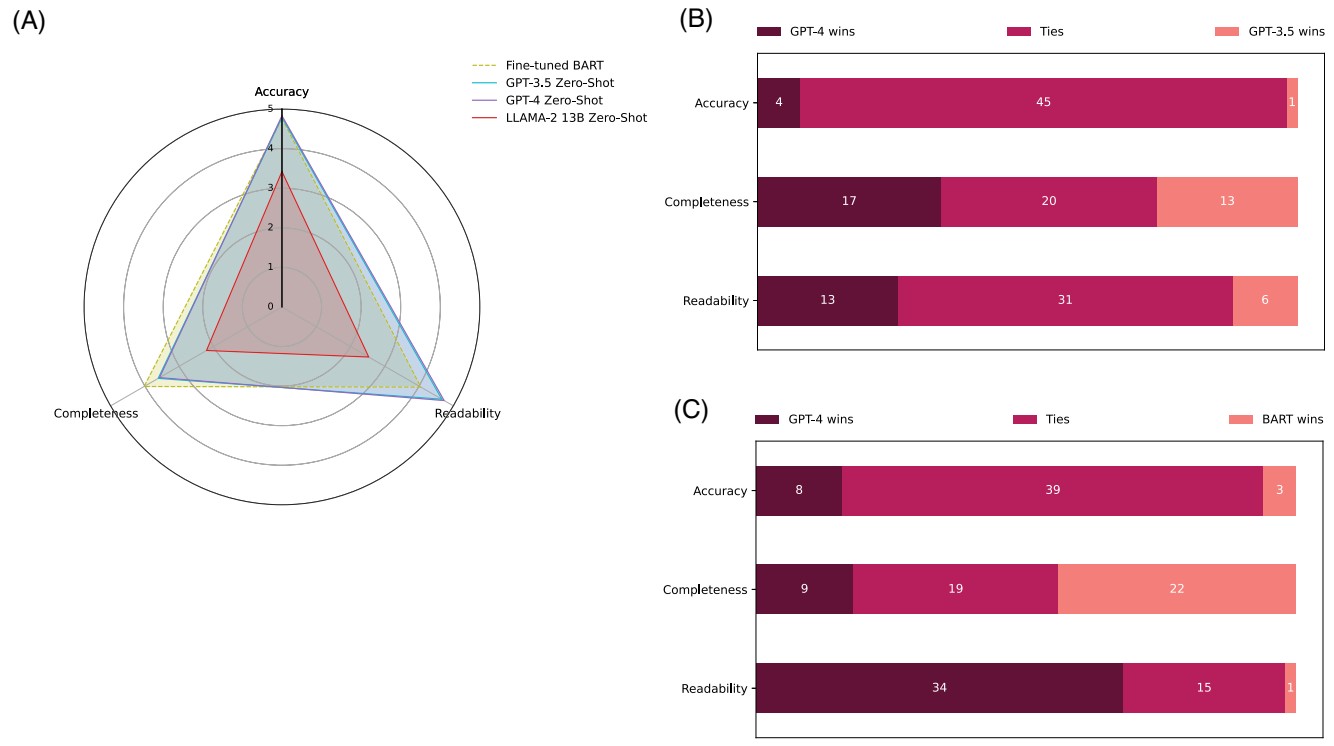

**Fig. 3 | Qualitative evaluation results on accuracy, completeness, and readability. A** The overall results of the fine-tuned BART, GPT-3.5 zero-shot, GPT-4 zero-shot, and LLaMA 2 zero-shot models on a scale of 1 to 5, based on random 50 testing instances from the PubMed Text Summarization dataset. **B** and **C** display the number of winning, tying, and losing cases when comparing GPT-4 zero-shot to GPT-3.5 zero-shot and GPT-4 zero-shot to the fine-tuned BART model, respectively. Table 4 shows the results in digits for complementary. Detailed results, including statistical tests and examples, are provided in Supplementary Information S3.

**Table 4 | Qualitative evaluation results on accuracy, completeness, and readability of the generated text for the fine-tuned BART, GPT-3.5 zero-shot, GPT-4 zero-shot, and LLaMA 2 zero-shot models on a scale of 1 to 5, based on random 50 testing instances from the PubMed Text Summarization dataset, to complement Fig. 3**

|              | Fine-tuned BART | GPT-3.5 zero-shot | GPT-4 zero-shot | LLaMA 2 zero-shot |
|--------------|-----------------|-------------------|-----------------|-------------------|
| Accuracy     | 4.76            | 4.79              | 4.83            | 3.42              |
| Completeness | 4.02            | 3.61              | 3.57            | 2.20              |
| Readability  | 4.05            | 3.57              | 4.73            | 2.53              |

accounting for up to 30% of the full testing set instances. Therefore, fine-tuning open-source LLMs for BioNLP tasks is still necessary to bridge the gap. Only through fine-tuning LLaMA 2, its overall performance is slightly higher than the one-shot GPT-4 (4%). However, it is worth noting that the model sizes of LLaMA 2 and PMC LLaMA are significantly smaller than those of GPT-3.5 and GPT-4, making it challenging to evaluate them on the same level. Additionally, open-source LLMs have the advantage of continued development and local deployment.

Another primary finding on open-source LLMs is that the results do not indicate significant performance improvement from continuously biomedical pre-trained LLMs (PMC LLaMA 13B vs. LLaMA 2 13B). As mentioned, our study reproduced similar results reported in PMC LLaMA 13B; however, we also found that directly fine-tuning LLaMA 2 yielded better or at least similar performance—and this is consistent across all 12 benchmarks. In the biomedical domain, representative foundation LLMs such as PMC LLaMA used 32 A100 GPUs[34], and Meditron used 128 A100 GPUs to continuously pretrain from LLaMA or LLaMA 2[49]. Our evaluation did not find significant performance improvement for PMC LLaMA; the Meditron study also

only reported ~3% improvement itself and only evaluated on question answering datasets. At a minimum, the results suggest the need for a more effective and sustainable approach to developing biomedical domain-specific LLMs.

The automatic metrics for text summarization and simplification tasks may not align with manual evaluations. As the quantitative results on text summarization and generation demonstrated, commonly used automatic evaluations such as Rouge, BERT, and BART scores consistently favored the fine-tuned BART's generated text, while manual evaluations show different results, indicating that GPT-3.5 and GPT-4 had competitive accuracy and much higher readability even under the zero-shot setting. Existing studies also reported that the automatic measures on LLM-generated text may not correlate to human preference[35,47]. The MS^2 benchmark used in the study also discussed the limitation of automatic measures, specifically for text summarization[50]. Additionally, the results highlight that completeness is a primary limitation when adapting GPT models to biomedical text generation tasks despite its competitive accuracy and readability scores.

Last, our evaluation on both performance and cost demonstrates a clear trade-off when using LLMs in practice. GPT-4 had the overall best performance in the 12 benchmarks, with an 8% improvement over GPT-3.5 but also at a higher cost (60 to 100 times higher than GPT-3.5). Notably, GPT-4 showed significantly higher performance, particularly in question-answering tasks that involve reasoning, such as over 20% improvement in MedQA compared to GPT-3.5. This observation is consistent with findings from other studies[27,38]. Note that newer versions of GPT-4, such as GPT-4 Turbo, may further reduce the cost of using GPT-4.

These findings lead to recommendations for downstream users to apply LLMs in BioNLP applications, summarized in Fig. 4. It provides suggestions on which BioNLP applications are recommended (or not) for LLMs, categorized by conditions (e.g., the zero/few-shot setting

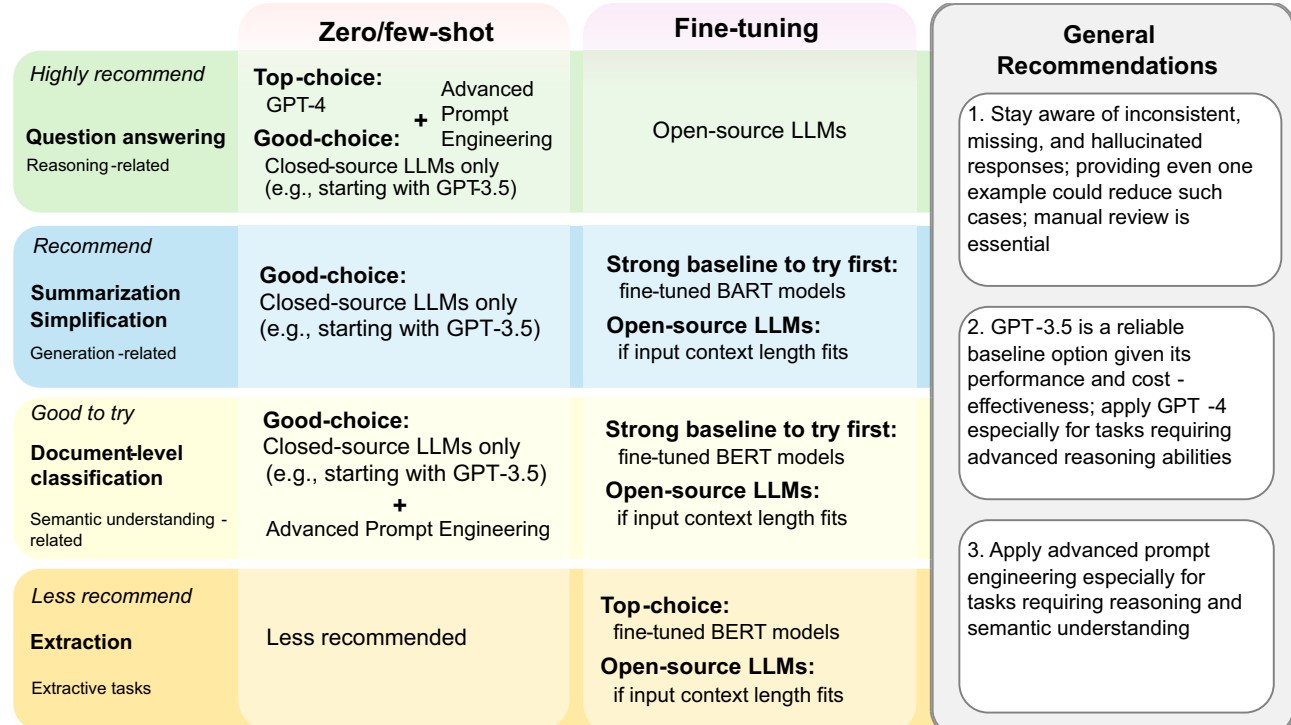

**Fig. 4 | Recommendations for using LLMs in BioNLP applications.** It presents specific task-based recommendations across different settings and offers general guidance on effectively applying LLMs in BioNLP.

when computational resources are limited) and additional tips (e.g., when advanced prompt engineering is more effective).

We also recognize the following two open problems and encourage a community effort for better usage of LLMs in BioNLP applications.

Adapting both data and evaluation paradigms is essential to maximize the benefits of LLMs in BioNLP applications. Arguably, the current datasets and evaluation settings in BioNLP are tailored to supervised (fine-tuning) methods and is not fair for LLMs. Those issues challenge the direct comparison between the fine-tuned biomedical domain-specific language models and zero/few shot of LLMs. The datasets for the tasks where LLMs excel are also limited in the biomedical domain. Further, the manual measures on biomedical text summarization also showed different results than that of all three automatic measures. These collectively suggest the current BioNLP evaluation frameworks have limitations when they are applied to LLMs[35,51]. They may not be able to accurately assess the full benefits of LLMs in biomedical applications, calling for the development of new evaluation datasets and methods for LLMs in bioNLP tasks.

Addressing inconsistencies, missingness, and hallucinations produced by LLMs is critical. The prevalence of inconsistencies, missingness, and hallucinations generated by LLMs is of concern, and we argue that they must be addressed for deployment. Our results demonstrate that providing just one shot could significantly reduce the occurrence of such issues, offering a simple solution. However, thorough examination in real-world scenario validations is still necessary. Additionally, more advanced approaches for validating LLMs' responses are expected for further improvement of their reliability and usability[47].

This study also has several limitations that should be acknowledged. While this study examined strong LLM representatives from each category (closed-source, open-source, and biomedical domain-specific), it is important to note that there are other LLMs, such as BARD[52] and Mistral[53], that have demonstrated strong performance in the literature. Additionally, while we investigated zero-shot, one-shot, dynamic K-nearest few-shot, and fine-tuning techniques, each of them

has variations, and there are also new approaches[54]. Given the rapidly growing nature of this area, our study cannot cover all of them. Instead, our aim is to establish baseline performance on the main BioNLP applications using commonly used LLMs and methods as representatives, and to make the datasets, methods, codes, and results publicly available. This enables downstream users to understand when and how to apply LLMs in their own use cases and to compare new LLMs and associated methods on the same benchmarks. In the future, we also plan to assess LLMs in real-world scenarios in the biomedical domain to further broaden the scope of the study.

## Methods
### Evaluation tasks, datasets, and metrics
Table 5 presents a summary of the evaluation tasks, datasets, and metrics. We benchmarked the models on the full testing sets of the twelve datasets from six BioNLP applications, which are BC5CDR-chemical and NCBI-disease for Named Entity Recognition, ChemProt and DDI2013 for relation extraction, HoC and LitCovid for multi-label document classification, and MedQA and PubMedQA for question answering, PubMed Text Summarization and MS^2 for text summarization, and Cochrane PLS and PLOS Text Simplification for text simplification. These datasets have been widely used in benchmarking biomedical text mining challenges[55-57] and evaluating biomedical language models[9-11,16]. The datasets are also available in the repository. We evaluated the datasets using the official evaluation metrics provided by the original dataset description papers, as well as commonly used metrics for method development or applications with the datasets, as documented in Table 5. Note that it is challenging to have a single one-size-fits-all metric, and some datasets and related studies used multiple evaluation metrics. Therefore, we also adopted secondary metrics for additional evaluations. A detailed description is below.

Named entity recognition. Named entity recognition is a task that involves identifying entities of interest from free text. The biomedical entities can be described in various ways, and resolving the ambiguities is crucial[58]. Named entity recognition is typically a sequence

**Table 5 | Evaluation datasets, dataset size, and evaluation metrics**

| | Training | Validation | Testing | Primary metrics | Secondary metrics |
|---|---|---|---|---|---|
| **Named entity recognition** | | | | | |
| BC5CDR-chemical[59] | 4560 | 4581 | 4797 | Entity-level F1[59,89] | |
| NCBI-disease[60] | 5424 | 923 | 940 | Entity-level F1[16,60] | |
| **Relation extraction** | | | | | |
| ChemProt[55] | 19,460 | 11,820 | 16,943 | Macro F1[90] | Micro F1[55,90] |
| DDI2013[62] | 18,779 | 7244 | 5761 | Macro F1[62,85] | Micro F1[16] |
| **Multi-label document classification** | | | | | |
| HoC[64] | 1108 | 157 | 315 | Macro F1[64,86] | Micro F1[86] |
| LitCovid[56] | 24,960 | 6239 | 2500 | Macro F1[56] | Micro F1[56] |
| **Question answering** | | | | | |
| MedQA 5-option[66] | 10,178 | 1272 | 1273 | Accuracy[66] | Macro F1[91] |
| PubMedQA[67] | 190,142 | 21,127 | 500 | Accuracy[67] | Macro F1[91] |
| **Text summarization** | | | | | |
| PubMed Text Summarization[a,68] | 117,108 | 6631 | 6658 | Rouge-L[68] | BERT Score[92], BART Score[93] |
| MS^2[b,50] | 14,188 | 2021 | - | Rouge-L[50] | BERT Score[94], BART Score[28] |
| **Text simplification** | | | | | |
| Cochrane PLS[69] | 3568 | 411 | 480 | Rouge-L[69] | FKGL[95], DCRS[96] |
| PLOS Text Simplification[70] | 26,124 | 1000 | 1000 | Rouge-L[70] | FKGL[70], DCRS[70] |

The related studies using the metrics are also provided. [a]We filtered the noisy instances with less than 50 words for the training and validation sets and kept the testing set untouched. [b]The gold standard of the testing set of MS^2 is not publicly available; we used the validation set instead.

labeling task, where each token is classified into a specific entity type. BC5CDR-chemical[59] and NCBI-disease[60] are manually annotated named entity recognition datasets for chemicals and diseases mentioned in biomedical literature, respectively. The exact match (that is, the predicted tokens must have the same text spans as the gold standard) F1-score was used to quantify the model performance.

Relation extraction. Relation extraction involves identifying the relationships between entities, which is important for drug repurposing and knowledge discovery[61]. Relation extraction is typically a multi-class classification problem, where a sentence or passage is given with identified entities and the goal is to classify the relation type between them. ChemProt[55] and DDI2013[62] are manually curated relation extraction datasets for protein-protein interactions and drug-drug interactions from biomedical literature, respectively. Macro and micro F1-scores were used to quantify the model performance.

Multi-label document classification. Multi-label document classification identifies semantic categories at the document-level. The semantic categories are effective for grasping the main topics and searching for relevant literature in the biomedical domain[63]. Unlike multi-class classification, which assigns only one label to an instance, multi-label classification can assign up to N labels to an instance. HoC[64] and LitCovid[56] are manually annotated multi-label document classification datasets for hallmarks of cancer (10 labels) and COVID-19 topics (7 labels), respectively. Macro and Micro F1 scores were used as the primary and secondary evaluation metrics, respectively.

Question answering. Question answering evaluates the knowledge and reasoning capabilities of a system in answering a given biomedical question with or without associated contexts[45]. Biomedical QA datasets such as MedQA and PubMedQA have been widely used in the evaluation of language models[65]. The MedQA dataset is collected from questions in the United States Medical License Examination (USMLE), where each instance contains a question (usually a patient description) and five answer choices (e.g., five potential diagnoses)[66]. The PubMedQA dataset includes biomedical research questions from PubMed, and the task is to use yes, no, or maybe to answer these questions with the corresponding abstracts[67]. Accuracy and macro F1-score are used as the primary and secondary evaluation metrics, respectively.

Text summarization. Text summarization produces a concise and coherent summary of a longer documents or multiple documents while preserving its essential content. We used two primary biomedical text summarization datasets: the PubMed text summarization benchmark[68] and MS^2[50]. The PubMed text summarization benchmark focuses on single document summarization where the input is a full PubMed article, and the gold standard output is its abstract. M2^2 in contrast, focuses on multi-document summarization where the input is a collection of PubMed articles, and the gold standard output is the abstract of a systematic review study that cites those articles. Both benchmarks used the ROUGE-L score as the primary evaluation metric; BERT score and BART score were used as secondary evaluation metrics.

Text simplification. Text simplification rephrases complex texts into simpler language while maintaining the original meaning, making the information more accessible to a broader audience. We used two primary biomedical text simplification datasets: Cochrane PLS[69] and the PLOS text simplification benchmark[70]. Cochrane PLS consists of the medical documents from the Cochrane Database of Systematic Reviews and the corresponding plain-language summary (PLS) written by the authors. The PLOS text simplification benchmark consists of articles from PLOS journals and the corresponding technical summary and PLS written by the authors. The ROUGE-L score was used as the primary evaluation metric. Flesch-Kincaid Grade Level (FKGL) and Dale-Chall Readability Score (DCRS), two commonly used evaluation metrics on readability[71] were used as the secondary evaluation metrics.

**Baselines**

For each dataset, we reported the reported SOTA fine-tuning result before the rise of LLMs as the baseline. The SOTA approaches involved fine-tuning (domain-specific) language models such as PubMedBERT[16], BioBERT[9], or BART[72] as the backbone. The fine-tuning still requires scalable manually labeled instances, which is challenging in the biomedical domain[32]. In contrast, LLMs may have the advantage when minimal manually labeled instances are available, and they do not require fine-tuning or retraining for every new task through zero/few-shot learning. Therefore, we used the existing SOTA results achieved

by the fine-tuning approaches to quantify the benefits and challenges of LLMs in BioNLP applications.

## Large language models

**Representative LLMs and their versions.** Both GPT-3.5 and GPT-4 have been regularly updated. For reproducibility, we used the snapshots gpt-3.5-turbo-16k-0613 and gpt-4-0613 for extractive tasks, and gpt-4-32k-0613 for generative tasks, considering their input and output token sizes. Regarding LLaMA 2, it is available in 7B, 13B, and 70B versions. We evaluated LLaMA 2 13B based on the computational resources required for fine-tuning, which is arguably the most common scenario applicable to BioNLP downstream applications. For PMC LLaMA, both 7B and 13B versions are available. Similarly, we used PMC LLaMA 13B, specifically evaluating it under the fine-tuning setting – the same setting used in its original study[34]. In the original study, PMC LLaMA was only evaluated on medical question answering tasks, combining multiple question answering datasets for fine-tuning. In our case, we fine-tuned each dataset separately and reported the results individually.

**Prompts.** To date, prompt design remains an open research problem[73–75]. We developed a prompt template that can be used across different tasks based on existing literature[74–77]. An annotated prompt example is provided in Supplementary Information S1 Prompt engineering, and we have made all the prompts publicly available in the repository. The prompt template contains (1) task descriptions (e.g., classifying relations), (2) input specifications (e.g., a sentence with labeled entities), (3) output specifications (e.g., the relation type), (4) task guidance (e.g., detailed descriptions or documentations on relation types), and (5) example demonstrations if examples from training sets are provided. This approach aligns with previous studies in the biomedical domain, which have demonstrated that incorporating task guidance into the prompt leads to improved performance[74,76] and was also employed and evaluated in our previous study, specifically focusing on named entity recognition[77]. We also adapted the SOTA example selection approach in the biomedical domain described below[27].

**Zero-shot and static few-shot.** We comparatively evaluated the zero-shot, one-shot, and five-shot learning performances. Only a few studies have made the selected examples available. For reproducibility and benchmarking, we first randomly selected the required number of examples in training sets, used the same selected examples for few-shot learning, and made the selected examples publicly available.

**Dynamic K-nearest few-shot.** In addition to zero- or static few-shot learning where fixed instructions are used for each instance, we further evaluated the LLMs under a dynamic few-shot learning setting. The dynamic few-shot learning is based on the MedPrompt approach, the SOTA method that demonstrated robust performance in medical question answering tasks without fine-tuning[27]. The essence is to use K training instances that are most similar to the test instance as the selected examples. We denote this setting as dynamic K-nearest few-shot, as the prompts for different test instances differ. Specifically, for each dataset, we used the SOTA text embedding model text-embedding-ada-002[54] to encode the instances and used cosine similarity as the metric for finding similar training instances to a testing instance. We tested dynamic K-nearest few-shot prompts with K equals to one, two, and five.

**Parameters for prompt engineering.** For zero-, one-, and few-shot approaches, we used a temperature parameter of 0 to minimize variance for both GPT and LLaMA-based models. Additionally, for LLaMA models, we maintained other parameters unchanged, set the maximum number of generated tokens per task, and truncated the instances due to the input length limit for the five-shot setting. Further details are provided in Supplementary Information S1 Prompt engineering, and the related codes are available in the repository.

**Fine-tuning.** We further conducted instruction fine-tuning on LLaMA 2 13B and PMC-LLaMA 13B. For each dataset, we fine-tuned LLaMA 2 13B and PMC- LLaMA 13B using its training set. The goal of instruction fine-tuning is defined by the objective function: $\text{argmax}_\theta \sum_{(x^i, y^i) \in (X, Y)} \log p(y^i | x^i; \theta)$, where $x^i$ represents the input instruction, $y^i$ is the ground truth response, and $\theta$ is the parameter set of the model. This function aims to maximize the likelihood of accurately predicting responses based on the given instructions. The fine-tuning is performed on eight H100 80G GPUs, over three epochs with a learning rate of 1e−5, a weight decay of 1e−5, a warmup ratio of 0.01, and Low-Rank Adaptation (LoRA) for parameter-effective tuning[78].

**Output parsing.** For extractive and classification tasks, we extracted the targeted predictions (e.g., classification types or multiple-choice options) from the raw outputs of LLMs with a combination of manual and automatic processing. We manually reviewed the processed outputs. Manual review showed that LLMs provided answers in inconsistent formats in some cases. For example, when presenting multiple-choice option C, the raw output examples included variations such as: "Based on the information provided, the most likely … is C. The thyroid gland is a common site for metastasis, and …", "Great! Let's go through the options. A. … B. …Therefore, the most likely diagnosis is C.", and "I'm happy to help! Based on the patient's symptoms and examination findings, … Therefore, option A is incorrect. …, so option D is incorrect. The correct answer is option C." (adapted from real responses with unnecessary details omitted). In such cases, automatic processing might overlook the answer, potentially lowering LLM accuracy. Thus, we manually extracted outputs in these instances to ensure fair credit. Additionally, we qualitatively evaluated the prevalence of such cases (providing responses in inconsistent formats), which will be introduced below.

## Evaluations

**Quantitative evaluations.** We summarized the evaluation metrics in Table 5 under zero-shot, static few-shot, dynamic K-nearest few-shot, and fine-tuning settings. The metrics are applicable to the entire testing sets of 12 datasets. We further conducted bootstrapping using a subsample size of 30 and repeated 100 times at a 95% confidence interval to report performance variance and performed a two-tailed Wilcoxon rank-sum test using SciPy[79]. Further details are provided in Supplementary Information S2 Quantitative evaluation results (S2.1. Result reporting).

**Qualitative evaluations on inconsistency, missing information, and hallucinations.** For the tasks where the gold standard is fixed, e.g., a classification type or multiple-choice option, we conducted qualitative evaluations on collectively hundreds of thousands of raw outputs of the LLMs (the raw outputs from three LLMs under zero- and one-shot conditions across three benchmarks) to categorize errors beyond inaccurate predictions. Specifically, we examined (1) inconsistent responses, where the responses are in different formats, (2) missingness, where the responses are missing, and (3) hallucinations, where LLMs fail to address the prompt and may contain repetitions and misinformation in the output[36]. We evaluated and reported the results in selected datasets: ChemProt, HoC, and MedQA.

**Qualitative evaluations on accuracy, completeness, and readability.** For the tasks with free-text gold standards, such as summaries, we conducted qualitative evaluations on the quality of generated text. Specifically, one senior resident and one junior resident evaluated four models: the fine-tuned BART model reported in the SOTA approach, GPT-3.5 zero-shot, GPT-4 zero-shot, and LLaMA 2 13B zero-shot on 50 random samples from the PubMed Text Summarization benchmark. Each annotator was provided with 600 annotations. To mitigate potential bias, the model outputs were all lowercased, their orders were randomly shuffled, and the annotators were unaware of the models being evaluated. They assessed three dimensions on a scale of 1−5: (1) accuracy, does the generated text contain correct information

from the original input, (2) completeness, does the generated text capture the key information from the original input, and (3) readability, is the generated text easy to read. The detailed evaluation guideline is provided in Supplementary Information S3 Qualitative evaluation on the PubMed Text Summarization Benchmark.

**Cost analysis.** We further conducted a cost analysis to quantify the trade-off between cost and accuracy when using GPT models. The cost of GPT models is determined by the number of input and output tokens. We tracked the tokens in the input prompts and output completions using the official model tokenizers provided by OpenAI (https://cookbook.openai.com/examples/how_to_count_tokens_with_tiktoken) and used the pricing table (https://azure.microsoft.com/en-us/pricing/details/cognitive-services/openai-service/) to compute the overall cost.

### Reporting summary
Further information on research design is available in the Nature Portfolio Reporting Summary linked to this article.

## Data availability
All data supporting the findings of this study, including source data, are available in the article and Supplementary Information, and can be accessed publicly via https://doi.org/10.5281/zenodo.14025500[37]. Additional data or requests for data can also be obtained from the corresponding authors upon request. Source data are provided with this paper.

## Code availability
The codes are publicly available via https://doi.org/10.5281/zenodo.14025500[37].

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

## Acknowledgements

This study is supported by the following National Institutes of Health grants: 1R01LM014604 (Q.C., R.A.A., and H.X), 4R00LM014024 (Q.C.), R01AG078154 (R.Z., and H.X), 1R01AG066749 (W.J.Z), W81XWH-22-1-0164 (W.J.Z), and the Intramural Research Program of the National Library of Medicine (Q.C., Q.J., P.L., Z.W., and Z.L).

## Author contributions

Q.C., Z.L., and H.X. designed the research. Q.C., Y.H., X.P., Q.X., Q.J., A.G., M.B.S., X.A., P.L., Z.W., V.K.K., K.P., J.H., H.H., F.L., and J.D. performed experiments and data analysis. Q.C., Z.L., and H.X. wrote and edited the manuscript. All authors contributed to the discussion and manuscript preparation.

## Funding

## Competing interests

Dr. Jingcheng Du and Dr. Hua Xu have research-related financial interests at Melax Technologies Inc. The remaining authors declare no competing interests.
