## [Transparent Peer Review file · Nature Communications]

Benchmarking large language models for biomedical natural language processing applications and recommendations

Corresponding Author: Dr Qingyu Chen

Version 0:

Reviewer comments:

Reviewer #1

(Remarks to the Author)

SUMMARY

This paper studies the performance of auto-regressive, decoder-based large language models (LLMs), as well as (as baselines) that of masked language modeling, encoder-based or encoder-decoder-based language models, on natural language processing tasks in the biomedical domain. Specifically, it evaluates three decoder-based LLMs (GPT 3.5, GPT 4; and in the supplementary material, the open-source LLaMA) on 12 datasets representing 7 tasks: named entity recognition (2 datasets), relation extraction (2 datasets), multi-label classification (2 datasets), semantic similarity (1 dataset), question answering (1 dataset), text summarization (2 datasets), text simplification (2 datasets). These tasks can be broadly categorized into generative (summarization and simplification) vs extractive (the remainder).

The performance of the LLMs is compared to that of the baseline models: one encoder-decoder-based language model (BART) for summarization and simplification and of one encoder-based language model (PubMedBERT) for the other tasks.

On top of the automatic evaluations, a number of manual evaluations are also performed and lead to complementary results.

Of note, the decoder-based LLMs are evaluated in zero-shot or one-shot mode, i.e., they are not specifically trained on the target tasks, whereas the other models are fine-tuned (i.e., specifically trained) on training datasets for each of the target tasks.

OVERALL COMMENTS

The obtained results are noteworthy:

The baseline models outperform the LLMs on 10 out of 12 datasets. The LLMs, although tested in zero- and one-shot modes, outperform the baseline models in 2 datasets (one text simplification and one question-answering) and obtain competitive performance in 4 more datasets. A manual evaluation of a 20-example sample of one of the two summarization datasets shows that humans give slightly higher scores to GPT-4 than to BART. The LLM errors detected by the automatic evaluation are categorized into Missing output, Inconsistent semantic output, Inconsistent syntactic output, and Artificial output.

These results, obtained on 12 datasets representing 7 tasks, are of importance: for this reviewer, they show that the now traditional encoder-based language models are still the best option when training data is available, whereas the much larger, decoder-based language models are better for some text generation tasks and may have useful performance in extractive tasks, all the more as they do not require specific training for these tasks.

Reading the paper also reveals a difficulty in assessing an LLM such as GPT-4: the high cost incurred to perform the experiments, which restricts the size of these experiments.

The conclusions of the paper, however, emphasize one aspect of the results, as detailed below. They should instead provide a more balanced view of the results.

The authors should assess the statistical significance of the observed differences, all the more as sample sizes are generally much smaller than in the original datasets. This also applies to the manual validation, performed on 20 examples only, which leads to rather small differences, as far as can be read from the graph in Fig. 4.A.

The code for the experiments is provided, and the datasets can be obtained too: this is a very good point that will facilitate replication.

DETAILED COMMENTS

The paper compares auto-regressive, decoder-based LLMs to encoder-[decoder-]based, masked language models on a variety of tasks. The first observation provided in the Results and discussion section is "Overall, the fine-tuned BERT/BART models achieved the highest performance." However, this key observation is down-graded or omitted everywhere else in the paper, including in the abstract, contributions, and conclusion, which put forward the second observation provided in the Results and discussion section: "However, the performance of GPT-4 was competitive in six out of the 12: it outperformed in two datasets [...] and had comparable performance in four datasets [...]". A simple count leads to the fact that the fine-tuned BERT/BART models obtained the best performance in 10 out of 12 datasets in that comparison. This should be the first result reported in the abstract and in the other passages of the paper that summarize the study results. Fair enough, the abstract later adds that "two major bottlenecks persist. Firstly, the zero- and few-shot performance of LLMs on extractive and classification tasks remains suboptimal, demonstrating up to a 30% difference compared to the fine-tuned BERT models." But whereas this is a major result of the study, it is presented as a second-level conclusion.

The decoder-based LLMs are tested in zero-shot and one-shot mode only, they are not specifically trained to perform the target tasks on the target datasets. The reason for doing so should be explained. This obviously creates a major difference in the operating mode of the two types of language models. While there are good reasons for doing so, these reasons should be exposed in the Introduction and Methods sections. Currently, the reader may only infer some of these motivations from a sentence in the Conclusion ("Given that zero- and few-shot learning of LLMs require minimal annotated data or computational effort to retrain models for specific tasks, [...]").

The authors use the category "semantic similarity and reasoning" to subsume the tasks associated with BIOSSES (semantic similarity) and PubMedQA (question-answering). This seems to imply that "reasoning" is intended to subsume question-answering. It is not obvious, however, that question-answering entails the use of reasoning, an ability that is more general and more diverse than what is needed to answer questions. This reviewer suggests dropping "reasoning" and directly using "question-answering": this will be clearer and will prevent the reader from erroneously interpreting the results as pertaining to a general reasoning capacity that would be examined in LLMs.

As mentioned by the authors, the choice of prompts for LLMs may strongly impact their results. The choice of the selected prompts is very quickly covered in the paper. How that prompt was chosen, whether others were tested, etc., should be explained in detail, together with references to work on prompt engineering.

The BIOSSES dataset, with 20 test examples, is very small, and it is unclear to what extent comparisons on such a small sample are meaningful.

Table 1 emphasizes the presence in the present work of an evaluation of generative tasks (summarization and simplification) and the presence of a manual evaluation: these are only performed in the present work, and one may wonder whether the need for a manual evaluation is correlated with the presence of generative tasks. This should be discussed. The authors may also want to cite a recent comparison of several language models on six question-answering datasets that also uses human evaluation:

Large language models encode clinical knowledge. *Nature*. 2023 Aug;620(7972):172-180. doi: 10.1038/s41586-023-06291-2. Epub 2023 Jul 12.

In Table 1, "Charge summary" should be "Discharge summary".

l172, the authors should specify whether the cost of GPT-4 is caused by compute cost or by access charges.

l189, "protein-protein iterations" -> interactions

l235, "that have been used as baselines of the benchmarks.": please specify by whom, i.e., in which papers.

l306, "examined 20 samples in NCBI-Disease": are these sentences or entities?
"which had the largest performance difference" between what and what conditions?

While it is commendable that the authors performed a manual inspection of errors, in the case of named entity recognition, if only one set of 20 entities was examined, this is a very small sample to obtain reliable conclusions.

More broadly, the number of instances in the four types of errors in the proposed taxonomy should be provided.

The caption of Figure 1 should mention "one-shot".

The relevance of the number of digits in the evaluation results of Table 4 (4 in most rows, 6 in the text simplification rows) should be justified with respect to the small number of selected test examples (200 in most datasets, 20 in BIOSSES)

In Figure 2, the authors should explain why, with 20 samples, the total number of errors is larger than 20.

Together with Figure 4, a complementary table would help read the actual values of the six scores displayed in Fig. 4.A.

In the supplementary material, one can notice that case and tokenization are not consistent across the provided summaries. The authors should say a word about whether or not this has consequences on evaluation results.

Reviewer #2

(Remarks to the Author)

This manuscript aims to conduct a comprehensive evaluation of LLMs (GPT3.5, GPT4) on biomedical NLP tasks. The authors evaluate these LLMs on 12 biomedical NLP datasets covering the tasks of NER, relation extraction, documentation classification, semantic similarity, summarization, and text simplification. Notably, this suite of datasets includes text generation tasks which have not been thoroughly evaluated for LLMs in this domain. The authors also conduct a manual validation on a subset of test samples with healthcare professionals on generation tasks. For extractive tasks, the authors categorize and analyze the types of errors produced by LLMs.

In terms of model performance, the authors report that zero- and few-shot prompting with GPT-4 is competitive or superior to fine-tuned BERT or BART models in 6 out of 12 benchmarking datasets, indicating their potential. While this work is not particularly novel, it is necessary and important for the community to have thorough evaluation of these new tools. However, towards this goal, there are a number of concerns and limitations that I hope the authors can address.

Main concerns:

- Limited LLMs evaluated: while the authors discuss using an open source LLM like Llama2, the results are not reported in the paper. While the same prompts may not work as well for Llama2, reasonable performance should be reachable with some moderate prompt engineering. The current work doesn't do due diligence comparing against open source LLMs. It is hard to claim a systematic evaluation of LLMs for biomedical NLP when only two closed API models from OpenAI are reported on.
- Evaluation datasets: the selection of main metrics reported per dataset is not well-justified. In many cases, they do not agree with the primary metrics reported by the task authors. For example, for PubmedQA, typically accuracy is reported where as the authors claim Macro-F1 in Table 2 and Pearson correlation in Table 4?? For text summarization, ROUGE-L and other metrics are usually reported, while the authors report ROUGE-1. For text simplification, it is also important to report metrics beyond Fleisch-Kincaid. These disagreements make it very difficult to compare these results with prior work.
- Variances/standard deviations are not reported for any finetuned or prompting experiments. How consistent are these results?
- It seems more appropriate to report previous finetuned SOTA results on all of the included datasets, rather than only the finetuned results on PubmedBERT and BART.
- One-shot ICL isn't standard. Typically, few-shot ICL incorporates at least a few examples (up to 5 or 10 is often seen in other work). While the authors include some limited experiments at higher example counts, their reasoning that more examples does not improve things is not correct. More shots improves things quite a bit for some of the models. Also, not all task types are represented in these experiments (e.g., no text generation tasks), so it is difficult to conclude anything. The selection strategy for ICL examples also matters, and is not discussed at all.
- The authors mentioned that LLMs can produce inconsistent output. How was this handled? Did the authors try to parse out the correct answer span to more fairly credit the LLMs when they are correct but have formatted the answer incorrectly?
- What temperature and decoding settings are used for the prompting experiments?
- Figure 2: how do LLM errors compare to errors of finetuned models?

Additional comments/questions:

- Table 1: What's the difference between qualitative and manual evaluation?
- Prior work concludes that automated evaluation metrics are not effective at capturing generated text quality for tasks like summarization and simplification, especially for LLMs and in specific tasks like MS². There should be some acknowledgement of this work when discussing human vs. automated evaluations for these tasks.

Reviewer #3

(Remarks to the Author)

This paper provides a comprehensive benchmarking of LLMs for BioNLP tasks, including information extraction, classification, and generation. Frontier LLMs such as GPT4 have demonstrated remarkable performance in both general and biomedical tasks. This study represents a valuable addition given its large-scale study over 12 datasets across 6 tasks. The study in its current form, however, still has some salient growth areas that need to be improved.

The most important growth area lies in the very limited few-shot settings adopted in evaluating GPT3.5 and GPT4. Specifically, the authors included at most one example (1-shot). This is likely too limited, esp. given the large context size frontier LLMs admit (32K tokens for GPT4, 128K for GPT4-Turbo). While the performance is already impressive as the authors observed, the limitation makes it hard to assess the true out-of-the-box potential. Evaluating five-shot performance for GPT4, as common in prior studies, would thus be important to shed light on such potential.

It's also well known that the selection of the few-shot examples and prompting might result in varying performance. Assessing such variance, e.g., through studies on the dev set, would be important to understand if the reported accuracies are indeed representative.

The authors also highlighted manual evaluation, which is indeed a very useful contribution. However, the presentation in the paper appears a bit misleading. The authors made it sound like manual inspection revealed additional failure modes for LLMs, whereas in effect, such errors were already accounted for in the quantitative evaluation for classification tasks (extract and document classification). So the manual evaluation is more like error analysis that reveals potential growth opportunities. In fact, the inconsistent output format might even signify that the reported results might be a lower bound estimate for LLM performance, if there is no sophisticated post-processing to extract answers from LLM output. LLM results may substantially improve, e.g., by postprocessing, or by better prompting that provides clearer instruction or examples for the output.

For generation tasks, on the other hand, the common concern is that the automatic metric (e.g., ROUGE) may not correlate well with human evaluation. This is especially important to assess as automatic metric is better suited to assess relative improvement for similar types of systems, but can be misleading when comparing different generation mechanisms, as shown in translation.

The authors might want to revise accordingly in various places that mentioned the manual evaluation, e.g., in Contributions section and the result sections.

Additional technical comments:

The evaluation of GPT4 was conducted by subsampling, which is concerning as that would make the comparison not head-to-head. The authors cited the reason as cost control, but this may be too important a limitation to ignore. Recently, GPT-4 Turbo has been released, which is 4-6 times cheaper than GPT4-32K, with an even larger context size (128K). It'd be important to conduct a head-to-head comparison, as well as going beyond 1-shot. Turbo should make these quite doable.

The paper mentioned 170 trillion as the parameter number for GPT4, which is a widely debunked estimate. The consensus is that it's unlikely to be many more than 1 trillion.

In discussing limitations of prior studies on LLMs in BioNLP, the authors nevertheless missed a wide array of well-known studies on GPT4 for biomedical applications. While these studies do not invalidate the significance of this proposed study, it is a glaring omission not to mention and discuss how they are related and different. E.g.:

Sparks of Artificial General Intelligence: Early experiments with GPT-4. Bubeck et al. Contains biomed examples.

The AI Revolution in Medicine: GPT-4 and Beyond. Lee et al. Contains extensive analysis of GPT4 in health applications.

Scaling Clinical Trial Matching Using Large Language Models: A Case Study in Oncology. Wong et al. MLHC 2023. Contains case study on GPT4 for structuring trial eligibility criteria for clinical trial matching.

Exploring the Boundaries of GPT-4 in Radiology. Liu et al. EMNLP 2023. Contains case study on GPT4 for structuring radiology reports.

Can Generalist Foundation Models Outcompete Special-Purpose Tuning? Case Study in Medicine. Nori, Lee, Zhang et al. Contains detailed analysis of prompt evaluation on GPT4 for medical QA evaluation.

Version 1:

Reviewer comments:

Reviewer #1

(Remarks to the Author)

Thank you for taking into account this reviewer's comments, adding more experiments, and extensively revising the paper. This represents a large amount of additional work that adds value to the paper.

This reviewer's main remaining concern involves the presentation of the main lessons learnt from the observed results. Besides, the presentation of the use of statistical significance tests needs small adjustments. (As in the authors responses, the line numbers refer to the version of the manuscript with highlighted revisions.)

1. The statements on the auto-regressive LLMs have been revised slightly, but the writing still gives the impression that the authors wish to praise these LLMs more than strictly necessary.

Based on the reported results on the comparison of LLMs to the state-of-the-art systems, an impartial observer would expect the main findings to be that the fine-tuned SOTA approaches still outperform zero- and few-shot LLMs in most tasks; only then, the specific list of tasks in which the LLMs outperform the SOTA or show competitive or promising performance should

be presented, with more detail provided on that. In contrast, the paper most often highlights the performance of LLMs first, only then presenting the better SOTA results in the background. Besides, even when acknowledging that the SOTA methods still outperform zero- and few-shot LLMs, the language used to express this still emphasizes the LLMs (they are the subjects of the verbs, they have superior performance) instead of the SOTA methods.

- The "Contributions of this study" (lines 195-199) first mention "We compared these models against fine-tuned BERT and BART models SOTA approaches mostly that fine-tuned (domain-specific) BERT or BART models, which have been well-established methods in BioNLP applications." They then report that "GPT-3.5 and GPT-4 *demonstrated superior zero- and few-shot performance for three types of BioNLP tasks*". This seems to mean that GPT-3.5 and GPT-4 obtain better results than BERT or BART models SOTA approaches for these three tasks. However, the quantitative evaluation results in Table 3 and Figure 1 report better results for SOTA approaches on all tasks but question answering. The text does continue (lines 199-203) with more detail in which the superiority of GPT-3.5 or GPT-4 is only asserted for question answering ("where they outperformed the state-of-the-art fine-tuning approaches"), but not for text summarization and simplification ("showing competitive accuracy and readability in generating text") nor document-level classification. This is globally misleading. Note that these contributions are presented before that on manual validation, which eventually highlight the better readability of GPT-4.0---but the better completeness of BART.

- This statement is further relaxed in the Abstract, which replaces the restrictive "for three types of" with a more absolute formulation (lines 45-6): "closed-source LLMs such as GPT-3.5 and GPT-4 *achieved superior zero- and few-shot performance,* " suggesting that this superiority happened on all datasets. In that context, the following text "*particularly in* (1) [...] (2) [...] and (3) [...]" implies that these are particularly strong areas, but does not convey the fact that GPT-3.5 and GPT-4 achieved inferior performance on the rest of the datasets.

- In that context, the statement (lines 50-4) "However, the overall performance of closed-source LLMs under zero- and few-shot settings is still *lower than the SOTA* performance: " comes as a surprise. The rest of the sentence is a potential source of further confusion since it states that "GPT-4 achieved the *highest* macro average of 0.4561 on the twelve datasets under the zero-shot setting" ("highest" seems to mean that GPT-4 obtains the top results), followed by "compared to 0.6531 for the SOTA." which eventually reveals that although GPT-4 is the "highest", it is below the SOTA---which is thus the true "highest".

- Lines 700-1 in the "Main findings and interpretations" state that "GPT-4 already outperformed previous fine-tuned SOTA approaches in MedQA and PubMedQA *with zero- and few-shot learning*", but for PubMedQA, the SOTA BioLinkBERT obtains 0.7340 whereas GPT-4 zero-shot learning obtains 0.6280. Therefore, "and" should be replaced with "or" (with zero-*or* few-shot learning), since zero-shot GPT-4 did not outperform BioLinkBERT on PubMedQA.

- In that section again, based on the reported result the main finding, which should be presented first, should be that the fine-tuned SOTA approaches still outperform zero- and few-shot LLMs in most tasks.

In the abstract and contributions, the use of the adjective "superior" might have an intended sense of "high-quality" rather than "better", but if this is the case, this reviewer considers it confusing and therefore misleading, especially in the context of a comparison with other models where the "better" sense is strongly expected.

2. Thank you for providing statistical significance test results. However, the text should specify between which results they are computed. Based upon the `quantitative_evaluation_statistic_test.xlsx` that the supplementary material points at, this reviewer assumes that these were conducted within each cell of Table 3 to test the significance of the difference between the top result of this cell and the other results of the same cell, but not across cells. First, such a kind of description should be provided in the main text. Second, the reader would like to know whether the top result in a row has a statistically significant difference to the next best results in the same row; in other words, whether it can be considered the best (for instance, one-shot often obtains higher results than zero-shot, but is the difference significant?). Ideally, this would include the comparison to the SOTA result.

3. Details

Line 191, "(4) semantic similarity and reasoning" remains and should be updated to question answering.

-> "We have (1) directly used "question answering" as suggested, (2) removed the BIOSSES given its limited size which was also suggested in a following comment, and (3) added MedQA as another question answering dataset for evaluation."

Lines 507-8, (0.6605 and 0.6707 in our study, respectively): This reviewer could not find these scores in Table 3.

(Remarks on code availability)

Reviewer #3

(Remarks to the Author)

The revision has addressed all concerns in my original review. Thanks authors for the detailed responses.

(Remarks on code availability)

Version 2:

Reviewer comments:

Reviewer #1

(Remarks to the Author)

I thank the authors for the new revision. It answers my last comments very satisfactorily.

1. The statements on the auto-regressive LLMs have been revised appropriately and now reflect the paper's findings very aptly.
2. Thank you for providing more detailed statistical significance test results. They are very helpful and give the reader a much clearer picture of the comparisons between different models.

Remaining typos:

Top of p.6: Remove spurious comma in "However, , closed-source LLMs"

p.13: Insert newline before heading "SOTA v.s. LLMs"
"v.s." should be "vs."

p.21: "As shown in Figure 3, GPT-4 achieved over a 0.7 F1-score with dynamic K-nearest shot": this should be "Figure 1"

(Remarks on code availability)

Responses to the reviewer comments

Overall major changes and new experiments

We highly appreciate the feedback from the reviewers and have addressed all the comments. The major changes and new experiments are summarized below. Point-by-point responses are further detailed per reviewer.

1. We have expanded our evaluation to include four Large Language Models (LLMs) by:
 - Incorporating LLaMA 2 13B (open-source) and PMC LLaMA 13B (domain-specific).
 - Introducing the dynamic K-nearest few-shot evaluation setting with K values of 1, 2, and 5.
 - Adding the fine-tuning evaluation setting.
2. We have quantitatively evaluated the full testing sets on all 12 benchmarks instead of using 200 subsamples per benchmark as previously done. This represents a 17-fold increase in scale. The models have been assessed on the full testing sets under zero-shot, one-shot, dynamic K-nearest-shot, and fine-tuning conditions, where applicable.
3. We have expanded the qualitative evaluation by reviewing hundreds of thousands of raw LLM outputs on the full testing sets to identify inconsistencies, missing information, and hallucinations.
4. We have further expanded qualitative evaluation on the assessments of accuracy, completeness, and readability of the text summarization task. Each annotator reviewed the four models on 50 random instances, resulting in a total of 600 annotations per annotator.
5. New content on GPT-3.5 and GPT-4 cost analysis has been added.
6. Recommendations have been provided to downstream users to guide them on when and how to use LLMs in their own use cases.
7. We have made datasets, codes, and results publicly available for downstream users to apply LLMs in their own use cases and to compare new LLMs and associated methods under the same settings.

The detailed point-by-point responses are provided below for each reviewer. Our responses are in blue.

Individual reviewer comments

Reviewer #1 (Remarks to the Author):

SUMMARY

This paper studies the performance of auto-regressive, decoder-based large language models (LLMs), as well as (as baselines) that of masked language modeling, encoder-based or encoder-decoder-based language models, on natural language processing tasks in the biomedical domain. Specifically, it evaluates three decoder-based LLMs (GPT 3.5, GPT 4; and in the supplementary material, the open-source LLaMA) on 12 datasets representing 7 tasks: named entity recognition (2 datasets), relation extraction (2 datasets), multi-label classification (2 datasets), semantic similarity (1 dataset), question answering (1 dataset), text summarization (2 datasets), text simplification (2 datasets). These tasks can be broadly categorized into generative (summarization and simplification) vs extractive (the remainder). The performance of the LLMs is compared to that of the baseline models: one encoder-decoder-based language model (BART) for summarization and simplification and of one encoder-based language model (PubMedBERT) for the other tasks.

On top of the automatic evaluations, a number of manual evaluations are also performed and lead to complementary results.

Of note, the decoder-based LLMs are evaluated in zero-shot or one-shot mode, i.e., they are not specifically trained on the target tasks, whereas the other models are fine-tuned (i.e., specifically trained) on training datasets for each of the target tasks.

OVERALL COMMENTS

The obtained results are noteworthy:

The baseline models outperform the LLMs on 10 out of 12 datasets. The LLMs, although tested in zero- and one-shot modes, outperform the baseline models in 2 datasets (one text simplification and one question-answering) and obtain competitive performance in 4 more datasets. A manual evaluation of a 20-example sample of one of the two summarization datasets shows that humans give slightly higher scores to GPT-4 than to BART. The LLM errors detected by the automatic evaluation are categorized into Missing output, Inconsistent semantic output, Inconsistent syntactic output, and Artificial output.

These results, obtained on 12 datasets representing 7 tasks, are of importance: for this reviewer, they show that the now traditional encoder-based language models are still the best option when training data is available, whereas the much larger, decoder-based language models are better for some text generation tasks and may have useful performance in extractive tasks, all the more as they do not require specific training for these tasks.

Reading the paper also reveals a difficulty in assessing an LLM such as GPT-4: the high cost incurred to perform the experiments, which restricts the size of these experiments.

The conclusions of the paper, however, emphasize one aspect of the results, as detailed below. They should instead provide a more balanced view of the results.

The authors should assess the statistical significance of the observed differences, all the more as sample sizes are generally much smaller than in the original datasets. This also applies to the manual validation, performed on 20 examples only, which leads to rather small differences, as far as can be read from the graph in Fig. 4.A.

The code for the experiments is provided, and the datasets can be obtained too: this is a very good point that will facilitate replication.

Response:

Thank you for the positive remarks, as well as the thoughtful comments. We have incorporated your suggestions and addressed the comments as shown below.

DETAILED COMMENTS

The paper compares auto-regressive, decoder-based LLMs to encoder-[decoder-]based, masked language models on a variety of tasks. The first observation provided in the Results and discussion section is "Overall, the fine-tuned BERT/BART models achieved the highest performance." However, this key observation is down-graded or omitted everywhere else in the paper, including in the abstract, contributions, and conclusion, which put forward the second observation provided in the Results and discussion section: "However, the performance of GPT-4 was competitive in six out of the 12: it outperformed in two datasets [...] and had comparable performance in four datasets [...]". A simple count leads to the fact that the fine-tuned BERT/BART models obtained the best performance in 10 out of 12 datasets in that comparison. This should be the first result reported in the abstract and in the other passages of the paper that summarize the study results. Fair enough, the abstract later adds that "two major bottlenecks persist. Firstly, the zero- and few-shot performance of LLMs on extractive and classification tasks remains suboptimal, demonstrating up to a 30% difference compared to the fine-tuned BERT models." But whereas this is a major result of the study, it is presented as a second-level conclusion.

Response:

Thanks for the comment. We have now provided a balanced discussion based on the updated experiments suggested by other reviewers. Specifically, we have evaluated four LLMs (GPT-3.5, GPT-4, LLaMA 2, and PMC LLaMA) on the full test sets of the 12 benchmarks (instead of 200 samples in the original manuscript) under zero-shot, static one-shot, dynamic K-nearest shot ($K = 1, 2, \text{ and } 5$), and fine-tuning. The LLMs were compared with the best reported SOTA performance in the literature.

GPT-4 remains the highest performance under the zero/few setting and the latest results also suggest closed-source LLMs have superior zero- and few-shot performance in (1) reasoning, outperforming the

SOTA performance in both question answering datasets, (2) text generation for summarization and simplification, where the manual evaluation showed they have a competitive accuracy and higher readability than the SOTA approach for text summarization, and (3) semantic understanding, whether they underperformed the SOTA approach but still have a reasonable performance. However, we also noted that the overall performance of closed-source LLMs under zero- and few-shot settings is still lower than the SOTA performance. GPT-4 achieved the highest macro average of 0.4561 on the twelve datasets under the zero-shot setting compared to 0.6531 for the SOTA approaches. The performance on extractive tasks remains suboptimal.

We present both sides in detail in the revised manuscript, specifically:

1. Lines 45 to 59, Page 2, where we present both potential and challenges results in the Abstract
2. Lines 198 to 211, Page 6, where we describe the results from both sides in the Introduction
3. Lines 511 to 532, Pages 15 to 16, where we describe the main findings in the Result
4. Lines 697 to 719, Page 23, where we discuss those findings and interpretations in detail in the Discussion

The decoder-based LLMs are tested in zero-shot and one-shot mode only, they are not specifically trained to perform the target tasks on the target datasets. The reason for doing so should be explained. This obviously creates a major difference in the operating mode of the two types of language models. While there are good reasons for doing so, these reasons should be exposed in the Introduction and Methods sections. Currently, the reader may only infer some of these motivations from a sentence in the Conclusion ("Given that zero- and few-shot learning of LLMs require minimal annotated data or computational effort to retrain models for specific tasks, [...]").

Response:

Thanks for the suggestions. We have now added the motivation in Lines 144 to 150, Page 5, in the Introduction and Lines 352 to 359, Page 10, in the Data and Methods. Basically, we explained fine-tuning approach still needs scalable manually annotated instances which is often challenging in the biomedical domain. LLMs have the potential to adapt in BioNLP research when minimal manually labeled instances are available, and they do not require fine-tuning or retraining for every new task through zero/few-shot learning.

The authors use the category "semantic similarity and reasoning" to subsume the tasks associated with BIOSSES (semantic similarity) and PubMedQA (question-answering). This seems to imply that "reasoning" is intended to subsume question-answering. It is not obvious, however, that question-answering entails the use of reasoning, an ability that is more general and more diverse than what is needed to answer questions. This reviewer suggests dropping "reasoning" and directly using "question-answering": this will be clearer and will prevent the reader from erroneously interpreting the results as pertaining to a general reasoning capacity that would be examined in LLMs.

Response:

Thanks for the suggestion. We have (1) directly used “question answering” as suggested, (2) removed the BIOSSES given its limited size which was also suggested in a following comment, and (3) added MedQA as another question answering dataset for evaluation.

As mentioned by the authors, the choice of prompts for LLMs may strongly impact their results. The choice of the selected prompts is very quickly covered in the paper. How that prompt was chosen, whether others were tested, etc., should be explained in detail, together with references to work on prompt engineering.

Response:

Thank you. We have now added the explanation per your comment. The prompt template was designed to be consistent with the existing studies, including task descriptions, input and output specifications, task documentation (we included it because it showed better performance in the biomedical domain according to the literature), and example demonstrations. We have also acknowledged that the prompt design is still an open and ongoing research topic. We further provided independent studies which reported their prompt engineering results on the specific datasets under the same setting. Our results are consistent with theirs.

The specific changes are:

1. Lines 389 to 399, Page 11, where we describe the prompt engineering in detail in the Data and Methods section
2. Lines 504 to 510, Page 15, where we provide independent studies which reported the results on selected datasets in our benchmarks under the same setting in the Results section

The BIOSSES dataset, with 20 test examples, is very small, and it is unclear to what extent comparisons on such a small sample are meaningful.

Response:

Thank you. Per your comment, we have removed the BIOSSES dataset and added MedQA instead.

Table 1 emphasizes the presence in the present work of an evaluation of generative tasks (summarization and simplification) and the presence of a manual evaluation: these are only performed in the present work, and one may wonder whether the need for a manual evaluation is correlated with the presence of generative tasks. This should be discussed.

The authors may also want to cite a recent comparison of several language models on six question-answering datasets that also uses human evaluation:

Large language models encode clinical knowledge. *Nature*. 2023 Aug;620(7972):172-180. doi: 10.1038/s41586-023-06291-2. Epub 2023 Jul 12.

Response:

Thank you. We have cited the study in Table 1 and added related discussions. We have also added early study results on LLMs and motivations on using LLMs in the biomedical domain suggested by Reviewer 3. The specific changes are Table 1 and Lines 132 to 143, Pages 4 to 5.

In Table 1, "Charge summary" should be "Discharge summary".

Response:

We have corrected it.

l172, the authors should specify whether the cost of GPT-4 is caused by compute cost or by access charges.

Response:

Thanks for the comment. The cost was based on the number of input tokens and output tokens multiplying by the unit price. To fully address your comment, we added a cost analysis component in the revised version. The specific changes are:

1. Lines 479 to 482, Page 14, where we described tracking the number of input tokens and output tokens and calculating the total cost in the Data and Methods section.
2. Figure 1, where we presented the cost of GPT-3.5 and GPT-4 on the full testing sets under zero-shot, one-shot, and few-shot in regard with their performance for all the 12 test sets in the Results section.
3. Lines 592 to 606, Page 18, where we described the trade-off between the performance and cost.

l189, "protein-protein iterations" -> interactions

Response:

Thanks. We have corrected it.

l235, "that have been used as baselines of the benchmarks.": please specify by whom, i.e., in which papers.

Response:

Thank you. We have now reported the SOTA approaches for each dataset as baselines instead of fine-tuning language models by ourselves as suggested by Reviewer 2. The results are directly extracted from the existing studies with related citations. The specific changes are:

1. Table 3, adding SOTA results and references
2. Lines 352 to 359, Page 10, describing using the SOTA approaches as baselines

Therefore, this sentence has been removed and no longer applicable.

l306, "examined 20 samples in NCBI-Disease": are these sentences or entities?

"which had the largest performance difference" between what and what conditions?

Response:

Thank you. Those sentences have been also removed and no longer applicable. The samples referred to sentences in the testing set; each sentence is a testing instance which may contain zero or many entities. For named entity recognition, as the official evaluation metric is entity-level F1-score, we categorized on the entity level instead of sentences: correct entities, wrong entities, missing entities, and entities with boundary issues.

We have significantly updated the experiment and analysis on the full testing sets suggested by your comment below and also from Reviewer 2. Specifically:

1. Lines 455 to 459, Page 13, where we updated the quantitative evaluation on the full testing sets
2. Lines 524 to 532, Pages 15 to 16, where we updated the results description on named entity recognition benchmarks
3. Figure 2(A), where we reported exact numbers of each category as suggested
4. Lines 624 to 643, Pages 20 to 21, where we discussed the error analysis on named entity recognition in detail

While it is commendable that the authors performed a manual inspection of errors, in the case of named entity recognition, if only one set of 20 entities was examined, this is a very small sample to obtain reliable conclusions.

Response:

Thank you. We have now analyzed the full testing set as mentioned in the response above.

More broadly, the number of instances in the four types of errors in the proposed taxonomy should be provided.

Response:

Thank you. We have now reported the number of instances in Figure 2(A) as mentioned in the response above.

The caption of Figure 1 should mention "one-shot".

Response:

Thank you. We have added "one-shot" to the caption. Figure 1 has now in the supplementary material S1.

The relevance of the number of digits in the evaluation results of Table 4 (4 in most rows, 6 in the text simplification rows) should be justified with respect to the small number of selected test examples (200 in most datasets, 20 in BIOSSES)

Response:

Thank you. Now the quantitative experiments and analysis have been performed on the full testing sets as suggested by Reviewer 2. We reported the dataset scale in Table 2. We have also removed BIOSSES based on your comment above.

In Figure 2, the authors should explain why, with 20 samples, the total number of errors is larger than 20.

Response:

Thank you. 20 samples are 20 sentences where each sentence contains zero or multiple entities. As the named entity recognition benchmarks evaluate entity-level F1-score, we reviewed every entity prediction (which could be more than the number of actual entities in the gold standard) and categorized them into (1) correct entities, where the predicted entities are correct with both text spans and entity types, (2) wrong entities, where the predicted entities are incorrect, (3) missing entities, where the true entities are not predicted, and (4) boundary issues, where the predicted entities are correct but with different text spans than the gold standard. As mentioned, we have made substantial changes on (1) doing the error analysis on the full testing set which contains 960 entities in the gold standard and (2) updating the descriptions and results in Lines 624 to 637, Lines 20 to 21.

Together with Figure 4, a complementary table would help read the actual values of the six scores displayed in Fig. 4.A.

Response:

Thank you. We have scaled the analysis to 50 random samples with four models based on the comments from Reviewer 2. The latest results are in Figure 3. We have added a complementary table Table 4 with the actual values as you suggested.

In the supplementary material, one can notice that case and tokenization are not consistent across the provided summaries. The authors should say a word about whether or not this has consequences on evaluation results.

Response:

Thank you again for your comment. During the evaluation, the order of the model outputs were randomly shuffled and the annotators were unaware of the models being evaluated. As mentioned above, now this evaluation scaled to four models and the annotators did not know their detail during the evaluation. We further lower the cases. We have updated the description in Lines 468 to 478, Lines 13 to 14.

Reviewer #2 (Remarks to the Author):

This manuscript aims to conduct a comprehensive evaluation of LLMs (GPT3.5, GPT4) on biomedical NLP tasks. The authors evaluate these LLMs on 12 biomedical NLP datasets covering the tasks of NER, relation extraction, documentation classification, semantic similarity, summarization, and text simplification. Notably, this suite of datasets includes text generation tasks which have not been thoroughly evaluated for LLMs in this domain. The authors also conduct a manual validation on a subset of test samples with healthcare professionals on generation tasks. For extractive tasks, the authors categorize and analyze the types of errors produced by LLMs.

In terms of model performance, the authors report that zero- and few-shot prompting with GPT-4 is competitive or superior to fine-tuned BERT or BART models in 6 out of 12 benchmarking datasets, indicating their potential. While this work is not particularly novel, it is necessary and important for the community to have thorough evaluation of these new tools. However, towards this goal, there are a number of concerns and limitations that I hope the authors can address.

Response:

We greatly appreciate the reviewer for acknowledging the necessity and importance of this research to the broader scientific community.

Main concerns:

- Limited LLMs evaluated: while the authors discuss using an open source LLM like Llama2, the results are not reported in the paper. While the same prompts may not work as well for Llama2, reasonable performance should be reachable with some moderate prompt engineering. The current work doesn't do due diligence comparing against open source LLMs. It is hard to claim a systematic evaluation of LLMs for biomedical NLP when only two closed API models from OpenAI are reported on.

Responses:

Thank you for this great suggestion. Per your comment, we have now evaluated four LLMs in total: GPT-3.5 and GPT-4 as the closed-source LLM representatives, LLaMA 2 13B as the open-source LLM representative, and PMC LLaMA 13B as the biomedical domain-specific LLM representative under zero-shot, static one-shot, dynamic K-nearest shot (K=1, 2, and 5), and fine-tuning settings on the full testing sets of the 12 datasets. Our evaluation covers quantitative evaluations, qualitative evaluations, and cost analysis.

Specifically:

1. Lines 27 to 35, Page 1, where we have updated the models, evaluation measures and settings in the Abstract.
2. Lines 185 to 187, Lines 193 to 197, and Lines 218 to 224, where we have described the models and evaluation approaches in the Introduction.
3. Lines 373 to 482 (Pages 12 to 14), where we have significantly revised and described the baselines, LLMs to evaluate, different evaluation settings, and manual evaluations in detail in the Data and Methods.

- Evaluation datasets: the selection of main metrics reported per dataset is not well-justified. In many cases, they do not agree with the primary metrics reported by the task authors. For example, for PubmedQA, typically accuracy is reported where as the authors claim Macro-F1 in Table 2 and Pearson correlation in Table 4?? For text summarization, ROUGE-L and other metrics are usually reported, while the authors report ROUGE-1. For text simplification, it is also important to report metrics beyond Fleisch-Kincaid. These disagreements make it very difficult to compare these results with prior work.

Response:

Thank you. We have significantly updated the evaluation metrics according to your comment. For each dataset, we now use the official evaluation metric if available as the primary metric. The primary metrics are also reported in the SOTA approach such that they can be directly compared. We also adopted other metrics if they are also commonly used for the task in the literature as secondary metrics and cited the related studies. For PubMedQA, we have now used Accuracy as the main metric. For text summarization benchmarks, we have now used ROUGE-L, BERT score, and BART score. For text simplification benchmarks, we have now used ROUGE-L, Flesch-Kincaid Grade Level, and Dale-Chall Readability Score.

Specifically,

1. Table 2, where we detailed evaluation metrics for each dataset with related references.
2. Lines 280 to 344, Pages 8 to 10, where we also updated the description of evaluation metrics for each task.

- Variances/standard deviations are not reported for any finetuned or prompting experiments. How consistent are these results?

Response:

Thank you. Due to the high cost of close-sourced LLMs such as GPT-4, we did not repeat the same experiment multiple times. For instance, as shown in Figure 1 and Lines 604 to 606, Page 18, the cost is around \$5,600 for GPT-4 five-shot on the PubMed Text Summarization dataset only; and our overall evaluation covered zero-shot, static one-shot, dynamic K-nearest shot ($K = 1, 2, \text{ and } 5$) on the entire 12 datasets. We also used the temperature of 0 to minimize the variance.

To address your concern, we (1) performed bootstrapping using a subsample size of 30 and repetition of 100 times at a 95% confidence interval, reported the performance variances, and conducted a two-tailed Wilcoxon rank-sum test to compare the methods; the changes are made in Lines 457 to 459, Table 3, and Supplementary Material S2 with performance variance and statistic test results fully available; and (2) compared with the independent studies that reported LLM-related performance on specific datasets in Lines 504 to 510, Page 15, where our reported results were consistent under the same setting.

- It seems more appropriate to report previous finetuned SOTA results on all of the included datasets, rather than only the finetuned results on PubmedBERT and BART.

Response:

Thanks. We have now used previous finetuned SOTA results for all the 12 datasets. We directly quote their reported performance on the full test sets and performed the evaluations of the LLMs on the entire test sets (in the original version, the evaluation was done on a 200 random sample for each dataset). Please also kindly note that many of the SOTA methods are not publicly available so we cannot perform bootstrapping and statistical tests.

The specific changes are:

1. Lines 352 to 359, Page 10, where we described using the SOTA results as the baseline.
2. Table 3, where we directly quoted the SOTA results and cited the references.

- One-shot ICL isn't standard. Typically, few-shot ICL incorporates at least a few examples (up to 5 or 10 is often seen in other work). While the authors include some limited experiments at higher example counts, their reasoning that more examples does not improve things is not correct. More shots improves things quite a bit for some of the models. Also, not all task types are represented in these experiments (e.g., no text generation tasks), so it is difficult to conclude anything. The selection strategy for ICL examples also matters, and is not discussed at all.

Response:

Thank you. We have now added dynamic K-nearest shot where $K = 1, 2, \text{ and } 5$ and evaluated this setting on all the twelve datasets. Reviewer 3 also suggested using five shots. We also have used zero-shot and static one-shot as baselines to compare the dynamic prompt and increasing number of shots. The dynamic K-nearest shot is adapted from MedPrompt, one of the best prompting techniques for the biomedical domain (Nori, H., et al, 2023. Can generalist foundation models outcompete special-purpose tuning? case study in medicine. arXiv preprint arXiv:2311.16452.). The essence is to use K training instances that are most similar to each test instance as the selected examples. We followed similar settings which used the state-of-the-art text embedding model text-embedding-ada-002 to encode the instances and used cosine similarity as the metric for finding similar training instances to a testing instance.

The new results are shown in Figure 1 and Supplementary Material S2. The results demonstrate that dynamic K-nearest shot is effective in question answering and document-level classification (e.g., dynamic one-nearest shot was higher than static one-shot; increasing the number of shot generally improved the performance); however, the improvement over named entity recognition, relation extraction, text summarization, and text simplification is not significant (in some tasks, dynamic five-nearest shot had lower performance than a static one-shot). We also demonstrate the cost for increasing the number of shot in Figure 1, as suggested by Reviewer 1. For both text summarization datasets, GPT-4 with five shots cost ~\$85 per 100 instances.

The specific changes are:

1. Lines 404 to 413, Lines 11 to 12, where we described the dynamic K-nearest few-shot approach in the Data and Methods

2. Figure 1, detailed results on the dynamic K-nearest few-shot approach (K = 1, 2, and 5) in comparison with zero-shot and static one-shot and associated costs on all the 12 datasets
3. Lines 568 to 591, Page 18, where we described and discussed the results of Figure 1 in detail in the Results section.

- The authors mentioned that LLMs can produce inconsistent output. How was this handled? Did the authors try to parse out the correct answer span to more fairly credit the LLMs when they are correct but have formatted the answer incorrectly?

Response:

Yes, we extracted the targeted predictions (e.g., classification types or multiple-choice options) from the raw outputs of LLMs with a combination of manual and automatic processing. We manually reviewed all the processed outputs. The processed outputs are then evaluated with gold standard. We have now added in Lines 479 to 483, Page 14, for clarification.

- What temperature and decoding settings are used for the prompting experiments?

Response:

We used temperature with 0 for both GPT and LLAMA based models. For LLAMA based models, additionally, we kept other parameters unchanged and set the maximum number of tokens per task. We have added detail in Lines 414 to 418, Page 12. The codes are also publicly available in the repository.

- Figure 2: how do LLM errors compare to errors of finetuned models?

Response:

Thanks for asking. As the SOTA model is not publicly available so we could not directly compare the errors. To fully address your comments, we used an alternative fine-tuned BioBERT model on NCBI Disease from an independent study, which had an entity-level F1-score of 0.8920 for comparison. It predicted 863 entities out of 960 correctly. Its wrong entities, missing entities, and boundary issues were 111, 97, and 269, respectively. We have added the descriptions in Lines 634 to 637, Page 21.

Additional comments/questions:

- Table 1: What's the difference between qualitative and manual evaluation?

Response:

Thank you for asking. Manual evaluation is part of the qualitative evaluations, which we have now revised it for clarification.

- Prior work concludes that automated evaluation metrics are not effective at capturing generated text

quality for tasks like summarization and simplification, especially for LLMs and in specific tasks like MS². There should be some acknowledgement of this work when discussing human vs. automated evaluations for these tasks.

Response:

Thank you again for the comments. We have acknowledged the study and added the discussion in Lines 748 to 749, Line 24.

Reviewer #3 (Remarks to the Author):

This paper provides a comprehensive benchmarking of LLMs for BioNLP tasks, including information extraction, classification, and generation. Frontier LLMs such as GPT4 have demonstrated remarkable performance in both general and biomedical tasks. This study represents a valuable addition given its large-scale study over 12 datasets across 6 tasks. The study in its current form, however, still has some salient growth areas that need to be improved.

Response:

Thank you for the positive feedback on the importance and comprehensiveness of this research. We have also incorporated your suggestions and addressed the comments below.

The most important growth area lies in the very limited few-shot settings adopted in evaluating GPT3.5 and GPT4. Specifically, the authors included at most one example (1-shot). This is likely too limited, esp. given the large context size frontier LLMs admit (32K tokens for GPT4, 128K for GPT4-Turbo). While the performance is already impressive as the authors observed, the limitation makes it hard to assess the true out-of-the-box potential. Evaluating five-shot performance for GPT4, as common in prior studies, would thus be important to shed light on such potential.

It's also well known that the selection of the few-shot examples and prompting might result in varying performance. Assessing such variance, e.g., through studies on the dev set, would be important to understand if the reported accuracies are indeed representative.

Response:

Thank you for the comments. Similar comments are also suggested by Reviewer 2. We have significantly revised the few-shot setting by incorporating your both suggestions. We have now added dynamic K-nearest shot where $K = 1, 2, \text{ and } 5$ and evaluated this setting on the full test set for each of the twelve datasets. The dynamic K-nearest shot is adapted from MedPrompt, one of the best prompting techniques for the biomedical domain (Nori, H., et al, 2023. Can generalist foundation models outcompete special-purpose tuning? case study in medicine. arXiv preprint arXiv:2311.16452.). The essence is to use K training instances that are most similar to each test instance as the selected examples. We followed similar settings which used the state-of-the-art text embedding model text-embedding-ada-002 to encode the instances and used cosine similarity as the metric for finding similar training instances to a testing instance. We also performed static zero-shot and one-shot as baselines. The results are shown in Figure 1 and Supplementary Material S2. As suggested by Reviewer 1, we also presented a cost analysis.

The specific changes are:

1. Lines 404 to 413, Lines 11 to 12, where we described the dynamic K-nearest few-shot approach in the Data and Methods
2. Figure 1, detailed results on the dynamic K-nearest few-shot approach ($K = 1, 2, \text{ and } 5$) in comparison with zero-shot and static one-shot and associated costs on all the 12 datasets
3. Lines 568 to 591, Page 18, where we described and discussed the results of Figure 1 in detail in the Results section.

The authors also highlighted manual evaluation, which is indeed a very useful contribution. However, the presentation in the paper appears a bit misleading. The authors made it sound like manual inspection revealed additional failure modes for LLMs, whereas in effect, such errors were already accounted for in the quantitative evaluation for classification tasks (extract and document classification). So the manual evaluation is more like error analysis that reveals potential growth opportunities. In fact, the inconsistent output format might even signify that the reported results might be a lower bound estimate for LLM performance, if there is no sophisticated post-processing to extract answers from LLM output. LLM results may substantially improve, e.g., by postprocessing, or by better prompting that provides clearer instruction or examples for the output.

Response:

Thank you for the comments and apologies for the confusion. We extracted the targeted predictions (e.g., classification types or multiple-choice options) from the raw outputs of LLMs with a combination of manual and automatic processing. We manually reviewed all the processed outputs. The processed outputs are then evaluated with gold standard. We documented inconsistent cases and other types for qualitative evaluation in addition. We have added a dedicated paragraph on Lines 427 to 439 on output parsing and provided examples on inconsistent cases where manual extraction is required.

For generation tasks, on the other hand, the common concern is that the automatic metric (e.g., ROUGE) may not correlate well with human evaluation. This is especially important to assess as automatic metric is better suited to assess relative improvement for similar types of systems, but can be misleading when comparing different generation mechanisms, as shown in translation. The authors might want to revise accordingly in various places that mentioned the manual evaluation, e.g., in Contributions section and the result sections.

Response:

Thanks. We have revised to specifically refer text summarization and simplification tasks instead of text generation throughout the manuscript. We have also added related discussion and studies mentioning the automatic metrics have relatively low correlation between human evaluation for text summarization and simplification tasks (Lines 747 to 749, Page 24).

Additional technical comments:

The evaluation of GPT4 was conducted by subsampling, which is concerning as that would make the comparison not head-to-head. The authors cited the reason as cost control, but this may be too important a limitation to ignore. Recently, GPT-4 Turbo has been released, which is 4-6 times cheaper than GPT4-32K, with an even larger context size (128K). It'd be important to conduct a head-to-head comparison, as well as going beyond 1-shot. Turbo should make these quite doable.

Response:

Thank you. We have used Azure to maintain HIPAA compliance; however, GPT-4 Turbo wasn't available on Azure during our revision. To fully address your comments, we have used GPT-4 and GPT4-32K and

have completed head-to-head comparisons on the full testing sets of all the 12 benchmarks and also go beyond 1-shot including dynamic few-shot learning as mentioned above.

The paper mentioned 170 trillion as the parameter number for GPT4, which is a widely debunked estimate. The consensus is that it's unlikely to be many more than 1 trillion.

Response:

Thank you. We have removed that reference and associated description.

In discussing limitations of prior studies on LLMs in BioNLP, the authors nevertheless missed a wide array of well-known studies on GPT4 for biomedical applications. While these studies do not invalidate the significance of this proposed study, it is a glaring omission not to mention and discuss how they are related and different. E.g.:

Sparks of Artificial General Intelligence: Early experiments with GPT-4. Bubeck et al. Contains biomed examples.

The AI Revolution in Medicine: GPT-4 and Beyond. Lee et al. Contains extensive analysis of GPT4 in health applications.

Scaling Clinical Trial Matching Using Large Language Models: A Case Study in Oncology. Wong et al. MLHC 2023. Contains case study on GPT4 for structuring trial eligibility criteria for clinical trial matching.

Exploring the Boundaries of GPT-4 in Radiology. Liu et al. EMNLP 2023. Contains case study on GPT4 for structuring radiology reports.

Can Generalist Foundation Models Outcompete Special-Purpose Tuning? Case Study in Medicine. Nori, Lee, Zhang et al. Contains detailed analysis of prompt evaluation on GPT4 for medical QA evaluation.

Response:

Thank you. We have included the studies and also recent reviews. We have also revised the discussion of the studies in Lines 132 to 143, Pages 4 to 5.

Responses to the comments

Overall major changes

We sincerely appreciate the comments from the reviewer and have addressed all of them. The major changes are summarized below.

- We have added the requested five-shot results to Table 3.
- We have significantly updated the representation of the fine-tuned SOTA results and related observations, especially for comparing the fine-tuned SOTA approaches with LLMs.
- We have also updated the statistical test representation.

The detailed line-by-line responses are provided below. Our responses are in blue. The line numbers are based on the tracked changes pdf document.

Reviewer #1:

Thank you for taking into account this reviewer's comments, adding more experiments, and extensively revising the paper. This represents a large amount of additional work that adds value to the paper.

Response:

Thank you for the positive feedback.

This reviewer's main remaining concern involves the presentation of the main lessons learnt from the observed results. Besides, the presentation of the use of statistical significance tests needs small adjustments. (As in the authors responses, the line numbers refer to the version of the manuscript with highlighted revisions.)

1. The statements on the auto-regressive LLMs have been revised slightly, but the writing still gives the impression that the authors wish to praise these LLMs more than strictly necessary.

Based on the reported results on the comparison of LLMs to the state-of-the-art systems, an impartial observer would expect the main findings to be that the fine-tuned SOTA approaches still outperform zero- and few-shot LLMs in most tasks; only then, the specific list of tasks in which the LLMs outperform the SOTA or show competitive or promising performance should be presented, with more detail provided on that. In contrast, the paper most often highlights the performance of LLMs first, only then presenting the better SOTA results in the background. Besides, even when acknowledging that the SOTA methods still outperform zero- and few-shot LLMs, the language used to express this still emphasizes the LLMs (they are the subjects of the verbs, they have superior performance) instead of the SOTA methods.

Response:

Thank you for the comments. We have emphasized that the SOTA fine-tuning approaches outperformed zero- and few-shot LLMs in most BioNLP applications as the main finding. We have described this finding first where possible and then detailed the LLM-related results. Additionally, we have made the SOTA fine-tuning approaches the subjects. Specifically:

Lines 41-61: We presented this main finding first and then detailed the LLM-related results in the Abstract.

Lines 187-205: We updated the main finding and related discussions in the Introduction.

Lines 447-483: We significantly revised the sub-section "SOTA vs. LLMs" in the Results.

Lines 646-655: We presented the main finding at the top of "Main Findings and Interpretations" in the Discussion.

We have also addressed your detailed comments related to this topic below.

- The "Contributions of this study" (lines 195-199) first mention "We compared these models against fine-tuned BERT and BART models SOTA approaches mostly that fine-tuned (domain-specific) BERT or BART models, which have been well-established methods in BioNLP applications." They then report that "GPT-3.5 and GPT-4 *demonstrated superior zero- and few-shot performance for three types of BioNLP tasks*". This seems to mean that GPT-3.5 and GPT-4 obtain better results than BERT or BART models SOTA approaches for these three tasks. However, the quantitative evaluation results in Table 3 and Figure 1 report better results for SOTA approaches on all tasks but question answering. The text does continue (lines 199-203) with more detail in which the superiority of GPT-3.5 or GPT-4 is only asserted for question answering ("where they outperformed the state-of-the-art fine-tuning approaches"), but not for text summarization and simplification ("showing competitive accuracy and readability in generating text") nor document-level classification. This is globally misleading. Note that these contributions are presented before that on manual validation, which eventually highlight the better readability of GPT-4.0--but the better completeness of BART.

Response:

We have significantly revised the flow and representation from Lines 181 to 196. First, we mentioned the SOTA fine-tuning approaches as baselines. Then, we presented the main finding that the SOTA fine-tuning approaches outperformed zero- and few-shot LLMs with a macro average of approximately 15% higher on the 12 benchmarks. We further specified that the closed-source LLMs had superior performance in reasoning tasks and noted that they had lower-than-SOTA but reasonable performance in the other two tasks.

- This statement is further relaxed in the Abstract, which replaces the restrictive "for three types of" with a more absolute formulation (lines 45-6): "closed-source LLMs such as GPT-3.5 and GPT-4 *achieved superior zero- and few-shot performance,* " suggesting that this superiority happened on all datasets. In that context, the following text "*particularly in* (1) [...] (2) [...] and (3) [...]" implies that these are particularly strong areas, but does not convey the fact that GPT-3.5 and GPT-4 achieved inferior performance on the rest of the datasets.

Response:

We have revised this part in the Abstract, Lines 41 to 51. We specified that the closed-source LLMs had better (replaced the word “superior” based on the comment below) zero- and few-shot performance in reasoning tasks. For the other two tasks, we noted that their performance was “lower-than-SOTA but reasonable.”

- In that context, the statement (lines 50-4) “However, the overall performance of closed-source LLMs under zero- and few-shot settings is still *lower than the SOTA* performance: ” comes as a surprise. The rest of the sentence is a potential source of further confusion since it states that “GPT-4 achieved the *highest* macro average of 0.4561 on the twelve datasets under the zero-shot setting” (“highest” seems to mean that GPT-4 obtains the top results), followed by “compared to 0.6531 for the SOTA.” which eventually reveals that although GPT-4 is the “highest”, it is below the SOTA---which is thus the true “highest”.

Response:

We have removed the sentence and updated Lines 41-44. We stated that the SOTA approaches had the best overall performance.

- Lines 700-1 in the “Main findings and interpretations” state that “GPT-4 already outperformed previous fine-tuned SOTA approaches in MedQA and PubMedQA *with zero- and few-shot learning*”, but for PubMedQA, the SOTA BioLinkBERT obtains 0.7340 whereas GPT-4 zero-shot learning obtains 0.6280. Therefore, “and” should be replaced with “or” (with zero- *or* few-shot learning), since zero-shot GPT-4 did not outperform BioLinkBERT on PubMedQA.

Response:

We have changed to “or” in Line 660.

- In that section again, based on the reported result the main finding, which should be presented first, should be that the fine-tuned SOTA approaches still outperform zero- and few-shot LLMs in most tasks.

Response:

We have revised Lines 646 to 655. We presented the main finding that the SOTA approach outperformed in most BioNLP applications first and detailed the largest performance difference in information extraction tasks.

In the abstract and contributions, the use of the adjective “superior” might have an intended sense of “high-quality” rather than “better”, but if this is the case, this reviewer considers it confusing and therefore misleading, especially in the context of a comparison with other models where the “better” sense is strongly expected.

Response:

We have used “better” instead of “superior” and limited it only to reasoning tasks.

2. Thank you for providing statistical significance test results. However, the text should specify between which results they are computed. Based upon the `quantitative_evaluation_statistic_test.xlsx` that the supplementary material points at, this reviewer assumes that these were conducted within each cell of Table 3 to test the significance of the difference between the top result of this cell and the other results of the same cell, but not across cells. First, such a kind of description should be provided in the main text. Second, the reader would like to know whether the top result in a row has a statistically significant difference to the next best results in the same row; in other words, whether it can be considered the best (for instance, one-shot often obtains higher results than zero-shot, but is the difference significant?). Ideally, this would include the comparison to the SOTA result.

Response:

Thank you for the comments. We have updated Table 3 Lines 408 to 426 and `quantitative_evaluation_statistic_test.xlsx`. As there are 11 model results in total under different settings in Table 3, we cannot show the significance of every pair of the results in the main table for presentation purpose. To address the comment, we categorize the results based on zero/few-shot and fine-tuning categories.

For each category, we performed the statistical tests on the model results from the same category and show the results with $p\text{-value} < 0.05$ only if it outperforms all other models under the same category. This will justify whether the top result from one-shot is significantly higher than the second-best result from zero-shot, as in the suggested example by the Reviewer. We have also provided the statistic results of every pair of the model results in `quantitative_evaluation_statistic_test.xlsx` for complement. Most of the SOTA models are not publicly available nor are the individual predictions per test instances, thus we cannot perform statistical tests but can only cross-reference their reported results in Table 3 instead.